# Demystifying Language Model Forgetting with Low-rank Example Associations

## Abstract

Large Language models (LLMs) suffer from forgetting of upstream data when fine-tuned. Despite efforts on mitigating forgetting, few have investigated whether, and how forgotten upstream examples are dependent on and associated with newly learned tasks. Insights on such associations enable efficient and targeted mitigation of forgetting. In this paper, we empirically analyze forgetting (measured in log-perplexity increase) that occurs in $N$ upstream examples of language modeling or instruction-tuning after fine-tuning LLMs on one of $M$ new tasks, visualized in $M \times N$ matrices. We demonstrate that the matrices display simple low-rank patterns, often well-approximated with multiplicative scalar effects of upstream examples and newly learned tasks. We also examine fine-grained associations with visualization and statistics. Leveraging the low-rank nature of the associations, we predict forgetting of upstream examples when fine-tuning on unseen tasks with matrix completion over the empirical associations. This enables fast identification of most forgotten examples without expensive inference on the entire upstream data. The approach, despite simplicity, outperforms prior approaches that learn semantic relationships of learned tasks and upstream examples with LMs for predicting forgetting. We demonstrate the practical utility of our analysis by showing statistically significantly reduced forgetting as we upweight predicted examples for replay at fine-tuning.

## 1 Introduction

There has been growing need for promptly updating LLMs to mitigate harmful behaviors, update outdated knowledge, and expand the set of application domains. Although fine-tuning allows efficient and incremental updates of models, it risks catastrophic forgetting (McCloskey & Cohen, 1989; Goodfellow et al., 2014) of upstream knowledge learned in the pre-training or instruction-tuning phase, causing unintended prediction changes over known information. This is problematic for the stability of online deployed LLM systems, limiting the feasibility of continual fine-tuning in practice (Raffel, 2023; Shi et al., 2024).

While extensive works have developed algorithms to mitigate forgetting (Wu et al., 2024), the increasing scale of the model and data necessitates more efficient and targeted approaches. A key aspect of the challenge is understanding when and what LLMs forget, prompting analysis into the patterns of frequently forgotten examples (Toneva et al., 2019; Maini et al., 2022; Zhang & Wu, 2024), and the impacts of models and hyperparameters on forgetting (Mirzadeh et al., 2022; Kalajdzievski, 2024; Que et al., 2024; Ibrahim et al., 2024). However, how the associations between learned tasks and upstream examples inform forgetting remains under-explored. Understanding such associations creates potential for targeted mitigation of forgetting over specific upstream examples when a new task is learned. While theoretical and empirical study indicate associations between learned and forgotten tasks in shallower neural networks (Lee et al., 2021; Goldfarb et al., 2024; Ramasesh et al., 2021), the associations are under-explored for LLMs, or measured regarding upstream data of language modeling or instruction-tuning.

In this paper, we empirically study the associations between learned tasks and forgotten upstream examples of language modeling or instruction-tuning. We experiment with OLMo-1B, OLMo-7B, OLMo-7B-Instruct (Groeneveld et al., 2024) and MPT-7B (Team, 2023) models where upstream data is open-source. We analyze forgetting (in log perplexity increase) over $N$ upstream examples, after

Figure 1: The problem setup of analyzing the associations between learned tasks and forgotten upstream examples as we fine-tune LLMs on one of unseen new tasks. Over total $N$ upstream examples and $M$ unseen tasks, we measure and record forgetting in a $M \times N$ matrix and analyze the statistical properties of the associations.

fine-tuning the model on one of $M$ unseen instruction-tuning tasks, and represent the results in a $M \times N$ matrix. Afterwards, we visualize the matrices and fit the observations with statistical models to analyze the associations. Figure 1 illustrates the analysis setup.

Our findings suggest that the associations between learned tasks and forgotten examples are usually simple and display a low-rank pattern, as simple regression models can decently fit of the associations. We extract more fine-grained associations with visualization and statistics, showing cases where specific upstream examples are forgotten when learning a new task. We demonstrate that forgetting does not correlate with many predefined similarity metrics of learned tasks and upstream examples, such as token overlap or gradient dot-products, making these metrics poor predictors of forgetting. Inspired by the findings, we directly utilize statistics of forgetting to predict example forgetting on unseen tasks by solving a matrix completion problem over the association matrices, analogical to collaborative filtering (Sarwar et al., 2001) in recommender systems, achieving both efficiency and interpretability. Our $k$-nearest neighbor (KNN) model outperforms prior approaches that learn semantic relations of two examples with LMs (Jin & Ren, 2024) . We verify the benefit of prediction by upweighting examples with higher predicted forgetting during replay as we fine-tune LLMs on new instruction-tuning tasks, achieving statistically significant improvement in alleviating forgetting compared to replaying random examples.

To summarize, the contributions of this paper are (1) an empirical analysis on how forgotten examples are associated with learned tasks in representative 1B and 7B language models, and (2) a novel approach of predicting example forgetting by solving a matrix completion problem over the empirical associations, and (3) a practical and efficient algorithm to mitigate forgetting during LLM fine-tuning by upweighting upstream examples for replay according to the predicted forgetting.

## 2 PROBLEM AND ANALYSIS SETUP

In this section, we start by formally defining the metrics of forgetting and set up the problem formulation of analyzing the associations between learned tasks and forgotten examples. We then introduce models and datasets used for collecting the statistics.

### 2.1 COLLECTING STATISTICS OF FORGETTING

**Upstream examples and learned tasks.** Large Language models (LLMs) are commonly pre-trained with language modeling objectives over a massive collection for corpora, with some LLMs further post-trained (instruction-tuned) to better follow human instructions. We use *upstream data* to refer to language modeling or instruction tuning training data used at the pre-training or post-training phase of LLMs. For upstream data of language modeling, we define each upstream example $x_j \in x_{1..N}$ as a chunk of document (*e.g.,* a Wikipedia article) of a model-specific maximum number of tokens. For instruction tuning, each $x_j \in x_{1..N}$ corresponds to a pair of instructions and ground truth responses.

**Measuring forgetting.** We fine-tune an LLM (or an instruction-tuned LLM) on one unseen instruction-tuning task $T_i$ from a collection of tasks $T_{1..M}$. This results in $M$ separately fine-tuned models $f_{1..M}$. We then evaluate performance degradation on each upstream example $x_i \in x_{1..N}$. We measure log perplexity as the performance metric as they are applicable to both language modeling

and instruction tuning, and are known to correlate well with other dataset-specific metrics (Hoffmann et al., 2022). For language modeling, log perplexity of an example $x_i$ is averaged over all tokens in the example; for instruction tuning, we average log perplexity over the ground truth response tokens only. We measure forgetting $z_{ij}$ that occurs on an upstream example $x_j \in x_{1..N}$ as increase (degradation) in log perplexity after fine-tuning the LM on a new task $T_i \in T_{1..M}$. We record forgetting $z_{ij}$ in an association matrix $Z \in \mathbb{R}^{M \times N}$.

## 2.2 Models and Datasets

Our analysis requires access to upstream data of LLMs for measuring their log perplexity changes occurred after fine-tuning. We experiment with OLMo-1B, OLMo-7B, OLMo-7B-Instruct, and MPT-7B where upstream data of language modeling and instruction-tuning is open-source.

**OLMo-1B and 7B.** OLMo models are pre-trained on Dolma (Soldaini et al., 2024), a massive collection of text covering diverse domains such as Wikipedia and news articles. We fine-tune LMs over 66 tasks from FLAN-V2 (Longpre et al., 2023), 11 tasks from Tulu V2 (Ivison et al., 2023), and 8 tasks from Dolly (Conover et al., 2023), obtaining 85 fine-tuned models. The collections cover diverse instruction-tuning tasks such as reading comprehension, math reasoning, and safety alignment. We then evaluate log perplexity increase on a 1% subset of Dolma-v1.6-Sample. Each upstream example is a maximum 2,048-token document from Dolma, resulting in 141,816 examples. For 7B models, We perform either full-parameter fine-tuning or LoRA adapter tuning (Hu et al., 2022) only, noted as OLMo-7B (Full FT) or OLMo-7B (LoRA) respectively.

**OLMo-7B-Instruct.** OLMo-7B-Instruct models are instruction-tuned on Tulu V2. In our experiments, we few-shot fine-tune OLMo-7B-Instruct over new task data from MMLU (Hendrycks et al., 2021), BBH (Suzgun et al., 2022), TruthfulQA (Lin et al., 2022), and Dolly, and evaluate log perplexity increase over a stratified sample of 10,718 examples from Tulu v2 as upstream examples. The number of training examples in these datasets are smaller (on the scale of tens to hundreds of examples), making the setup closer to LLM error editing (Yao et al., 2023; Zhang et al., 2024). We perform either full-parameter fine-tuning or LoRA fine-tuning.

**MPT-7B**. MPT models are pre-trained on a diverse collection of corpora. We fine-tune the LM over the same 85 instruction-tuning tasks as OLMo models and evaluate forgetting on Redpajama (Computer, 2023), involved as a part of the pretraining corpora of MPT. We sample 10,000 maximum 2,048 token documents from Redpajama as upstream examples.

We use a learning rate of $2e^{-6}$ for full-parameter fine-tuning and $1e^{-4}$ for LoRA fine-tuning. We include the other training details in Appendix A.

# 3 Associations between Learned Tasks and Forgotten Examples

In this section, we address the research questions regarding the associations between learned tasks and forgotten upstream examples represented in the $M \times N$ association matrices $Z$. We start by examining how complicated are the associations represented by the matrices $Z$ (Sec. 3.1). We then extract and analyze more fine-grained associations in $Z$ (Sec. 3.2), and examine how the associations correlate with similarity measures of learned tasks and upstream examples (Sec.3.3).

## 3.1 How Complicated are the Associations?

In a hypothetical extreme case, the associations between the learned tasks and forgotten examples can be very simple, *e.g.*, where upstream examples are forgotten regardless of learned tasks; or it can be highly complicated, *e.g.,* where different new tasks cause very specific subsets of upstream examples to be forgotten. Figure 1 (right) illustrates how such simple or complicated associations may display in visualized matrices. To understand the actual associations, we start by visualizing the association matrix $Z$ collected in the setups described in Sec. 2.2, and perform quantitative analysis as we examine whether simple regression models can fit $Z$ with low error.

**Analysis on the visualized association matrices.** We visualize the association matrices $Z$ in Figure 2 for OLMo-1B, OLMo-7B, OLMo-7B-Instruct as we perform full-parameter fine-tuning. The matrices of MPT and LoRA fine-tuning of OLMo-7B and OLMo-7B-Instruct are presented

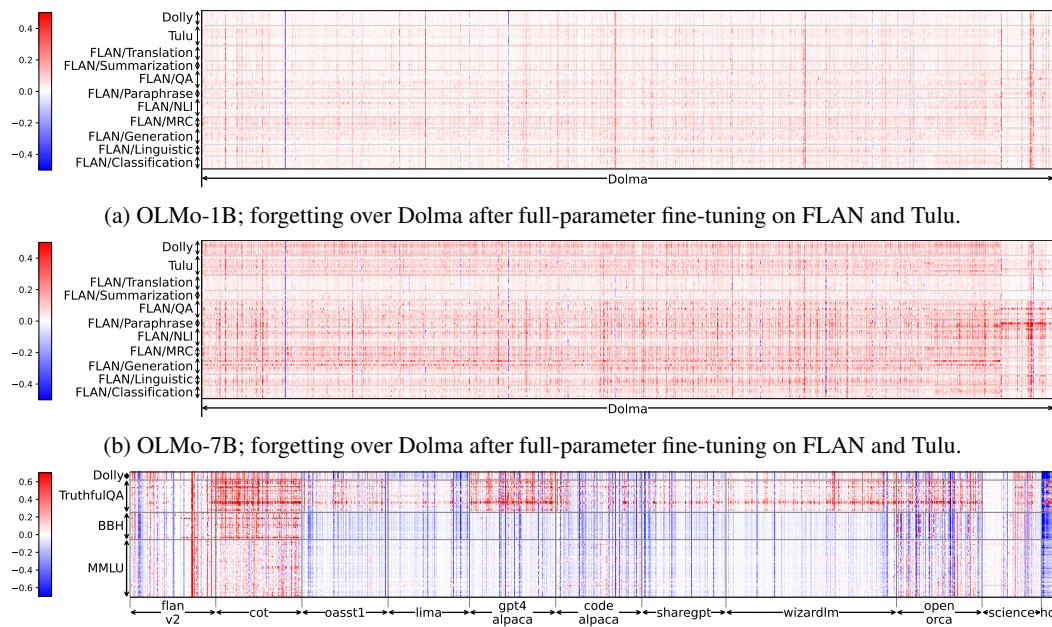

(a) OLMo-1B; forgetting over Dolma after full-parameter fine-tuning on FLAN and Tulu.

(b) OLMo-7B; forgetting over Dolma after full-parameter fine-tuning on FLAN and Tulu.

(c) OLMo-7B-Instruct; forgetting over Tulu after full-parameter fine-tuning on unseen instruction-tuning tasks.

Figure 2: Visualized matrices of associations between learned tasks and forgotten examples. We plot forgetting (log-perplexity increase) that occurs on an upstream example (in $x$-axis) after learning a new task (in $y$-axis). Log-perplexity increase can be zero or negative, which implies no forgetting.

in Figure 7 in Appendix. We see $Z$ generally displays a neat and simple pattern. We notice that certain upstream are more prone to forgetting, while some are never forgotten (displayed as all-zero columns). Nevertheless, in some cases, upstream examples that are never forgotten elsewhere are forgotten by learning specific new tasks (*e.g.* wizardlm examples are almost only forgotten after learning tasks from TruthfulQA in Figure 2(c)). It implies the association is a mixture of simple and more complicated patterns.

**Quantitative evaluation of simplicity of the associations**. We quantitatively measure how well the association matrices $Z$ can be approximated with simple regression models with a small number of learnable parameters. We consider (1) additive linear models, where $z_{ij} = b + \alpha_i + \beta_j + \epsilon$, where $\alpha_i$ and $\beta_j$ are learnable parameters associated with each new task or upstream example. (2) multiplicative models (SVD with rank $r$=1), where $z_{ij} = s\alpha_i\beta_j + \epsilon$. Both models involve $M + N$ learnable parameters plus a bias term. We then measure $R^2$ as the metrics of determining how well the regression models fit the association matrices $Z$. Let $f_{ij}$ be the fitted value, $R^2$ is defined as $1 - \sum_{i,j}(z_{ij} - f_{ij})^2 / \sum_{i,j}(z_{ij} - \bar{Z})^2$. $R^2$ ranges between 0 and 1, indicating the portion of total variance in $Z$ that is explained by the regression models. We report $R^2$ of additive and multiplicative models under different setups in Figure 3 (a).

**Multiplicative models fit the associations better.** In 5 out of 6 setups presented in Figure 3 (a), the multiplicative model achieves a better fit than additive models at the same number of trainable parameters. An interpretation of the multiplicative models is that each upstream example is associated with their tendency of being forgotten ($\beta_j$); the learned tasks trivially determine how fast all upstream examples are forgotten (with $\alpha_i$). This also creates cases where certain upstream are never forgotten ($\beta_j \approx 0$), or learn certain tasks causing little forgetting on any upstream examples ($\alpha_i \approx 0$).

**A large portion of the variance in the association can be explained by the multiplicative model.** We notice the multiplicative models achieve $R^2$ between 0.43 to 0.72 in different setups despite their simplicity, suggesting generally simple associations between learned tasks and forgotten examples. This finding is emphasized by the diversity of the learned tasks we considered (spanning from reading comprehension, coding, to safety alignment) and the broad coverage of domains in upstream language modeling and instruction-tuning data. Nevertheless, we also notice the $R^2$ scores are relatively lower

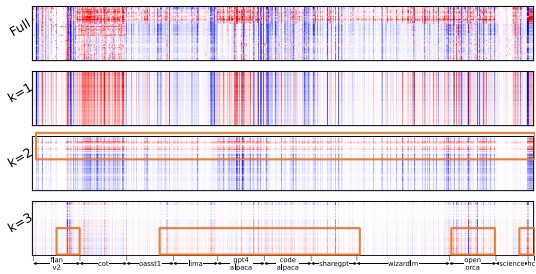

(a) Fitting $Z$ with additive or multiplicative models   (b) Fitting $Z$ with progressively higher rank matrices

Figure 3: $R^2$ of fitting the associations between learned tasks and forgotten examples with (a) simple additive linear or multiplicative models, or (b) progressively more expressive models.

(0.4 to 0.5) on OLMo-7B and OLMo-7B-Instruct, implying more complicated associations than the other setups (OLMo-1B, MPT-7B, and LoRA fine-tuning of OLMo-7B models), which elicits our next research question.

## 3.2 WHAT ARE MORE FINE-GRAINED ASSOCIATIONS?

We extract more fine-grained associations where fine-tuning on a task causes a specific set of upstream examples to be forgotten. We perform Singular Value Decomposition (SVD) over the association matrices $Z$ and progressively reconstruct $Z$ with up to $r$-th singular values and vectors as $Z_r = \sum_{k=1}^{r} s_k \boldsymbol{\alpha}_k \boldsymbol{\beta}_k^T$. $Z_r$ is also the optimal rank-$r$ matrix that minimizes the Frobenius norm $||Z - Z_r||_F$ and the $R^2$ score according to the property of SVD. The first component $Z_1$ corresponds to the multiplicative model we examined in Sec. 3.1; the rest of components $Z_{2..M}$ captures fine-grained associations in $Z$.

**SVD reveals fine-grained associations**. We present $R^2$ of fitting the assoication with $Z_r$ with progressively larger rank $r$ in Figure 3 (b). We notice that for OLMo-7B and OLMo-7B-Instruct (full-parameter fine-tuning), $R^2$ quickly increases to 0.69 and 0.78 with $Z_3$ (reconstruction with up to the third component). We visual-

Figure 4: Patterns in associations captured by $k$-th component in SVD of the assoication matrix $Z$, exemplified with OLMo-7B-Instruct (full-parameter tuning, also in Fig. 2(c)). The highlighted regions display patterns that TruthfulQA tasks cause more forgetting ($k = 2$) and MMLU causes more forgetting on certain tasks in FLAN v2 and Open-Orca ($k = 3$).

ize the matrices reconstructed from each component $s_k \boldsymbol{\alpha}_k \boldsymbol{\beta}_k^T$ in SVD of $Z$. Figure 4 provides an example of patterns captured by the $k$-th component in OLMo-7B-Instruct. We demonstrate how the visualization extracts patterns of forgetting that is conditional on the learned tasks. In Appendix D, we further examine interpretable patterns in the decomposition of $Z$ in OLMo-1B and 7B.

## 3.3 HOW DO THE SIMILARITY MEASURES INTERPRET THE ASSOCIATIONS?

We examine whether the associations between learned tasks and forgotten examples are interpretable from the similarity between the learned tasks and upstream examples. We consider (1) heuristic similarity measures, such as textual similarity, and (2) theoretically grounded approximations, such as inner products of gradients.

**Textual similarity.** We measure textual cosine similarity $z_{ij}^{\text{text}}$ between learned tasks and forgotten examples with TF-IDF vectorized features over each pair of learned tasks $T_i$ and upstream examples $x_j$. We also measure text representation similarity with final layer representations of OLMo-1B.

**Inner products between projected gradients and model weight updates**. The increase of the log perplexity $z_{ij}$ (also known as the cross-entropy loss) can be approximated with inner products $z_{ij}^{\text{g-w}} = \langle \nabla_\theta f(x_j), \theta_{T_i} - \theta_0 \rangle$ under first-order Taylor expansion (Lee et al., 2019; Doan et al., 2020), where $\nabla_\theta f(x_j)$ is the gradient of the loss of $x_j$ at the initial model before fine-tuning, and $\theta_{T_i} - \theta_0$

are the updates in the model weights after fine-tuning. Following Park et al. (2023); Xia et al. (2024), we use a random projection matrix $\boldsymbol{P} \sim \mathcal{N}_{|\theta| \times d}(0, 1)$ to reduce the dimension of the gradients or the weight changes to save the cost of storing pre-computed statistics, which preserves the inner products with high probability (Johnson & Lindenstrauss, 1984).

**Inner products between projected gradients**. We also measure the negative inner products of the loss gradients between the upstream example $x_j$ and a learned task $T_i$, $z_{ij}^{\text{g-g}} = -\langle \nabla_\theta f(x_j), \nabla_\theta f(T_i) \rangle$, as an approximation of forgetting (Lopez-Paz & Ranzato, 2017; Chaudhry et al., 2019).

**Forgetting correlates poorly with similarity measures of learned tasks and upstream examples.** We evaluate correlations between the actual forgetting $z_{ij}$ and the various similarity measures $\{z_{ij}^{\text{text}}, z_{ij}^{\text{g-w}}, z_{ij}^{\text{g-g}}\}$ on OLMo-1B and summarize the results in Table 1. We notice that none of the similarity measures correlates with the actual forgetting, with a correlation $|\rho| < 0.1$. An interpretation of the low correlation is that the model weights deviate from the initial weights after fine-tuning to a region where first-order approximation of forgetting

Table 1: Correlations between various measures of similarity and the actual forgetting on upstream examples after fine-tuning on a task.

|  | Pearson $\rho$ | Spearman $\rho$ |
| --- | --- | --- |
| Textual (TF-IDF) | -0.049 | -0.035 |
| Textual (Representation) | 0.021 | 0.017 |
| Gradient-Weight Diff. $z^{\text{g-w}}$ | -0.003 | -0.009 |
| Gradient-Gradient Diff. $z^{\text{g-g}}$ | 0.061 | 0.052 |

does not hold. We further visualize the matrices of $z_{ij}^{\text{g-w}}$ and $z_{ij}^{\text{g-g}}$ in Fig. 8 in Appendix and provide a side-by-side comparison with the matrices of forgetting $Z$. The visualization displays distinct patterns among the three matrices. These results imply that although the association matrices $Z$ display a simple pattern, they are not well-interpreted with common similarity measures of learned tasks and forgotten examples.

## 4 PREDICTING EXAMPLE FORGETTING WITH ASSOCIATION MATRIX COMPLETION

We utilize our findings in Sec. 3 to predict example forgetting as the model learns a new task, a problem also studied in prior works (Jin & Ren, 2024). Finding out the most forgotten examples allows better spot of the behavior changes of the models, enabling efficient and targeted approaches to mitigate forgetting, *e.g.,* by replaying these examples (Aljundi et al., 2019a; Wang et al., 2024). Although the ground truth forgetting can be directly obtained by running inference with the fine-tuned model over the upstream data, this requires extensive computational resources. We restrict the prediction methods to be computationally efficient.

Our analysis in Sec. 3 suggests that (1) the associations between learned tasks and forgotten examples display simple statistical patterns, while (2) the associations correlate poorly with many similarity metrics. Therefore, we hypothesize that leveraging the statistics of forgetting allows more effective prediction of forgetting than leveraging the contents of the tasks and examples. Following this intuition, we formulate prediction of example forgetting as a matrix completion problem over the empirical associations $Z$, analogical to collaborative filtering in recommender systems (Sarwar et al., 2001), where scalable approaches are studied extensively. We start by setting up the problem formulation of predicting example forgetting, and evaluate the performance of prediction of different approaches. We present reduced forgetting by utilizing the prediction outcomes during fine-tuning.

### 4.1 TRAINING AND EVALUATION OF FORGETTING PREDICTION

Our goal is to accurately predict forgetting $z_{ij}$ over upstream examples $x_{1..M}$ when the model is fine-tuned on an unseen task $T_j$ with a prediction model $g$, without running expensive LLM inference on all $x_{1..M}$. To evaluate this, we create training and test splits by partitioning the set of fine-tuning tasks (noted as $\mathcal{T}_{\text{train}}$ and $\mathcal{T}_{\text{test}}$) and the rows of the association matrices $Z$. We further control whether the $\mathcal{T}_{\text{train}}$ and $\mathcal{T}_{\text{test}}$ are from the same category of the tasks to test both in-domain and out-of-domain generalization ability of the prediction models. For OLMo-1B and 7B experiments, we use FLAN and the in-domain tasks and Tulu and Dolly as out-of-domain testing tasks. For OLMo-7B-Instruct experiments, we use MMLU and BBH as in-domain tasks and use TruthfulQA and Dolly as out-

Table 2: RMSE of predicting example forgetting over a held-out set of upstream examples after fine-tuning LMs on unseen new tasks. We report averaged performance over different seed sets ($\mathcal{S}$) of upstream examples with known ground truth forgetting beforehand.

|  | In-Domain | | | | | | Out-of-Domain | | | | | |
|---|---|---|---|---|---|---|---|---|---|---|---|---|
|  | OLMo-1B | OLMo-7B | | MPT-7B | OLMo-7B-Inst. | | OLMo-1B | OLMo-7B | | MPT | OLMo-7B-Inst. | |
|  | Full FT | Full FT | LoRA | Full FT | Full FT | LoRA | Full FT | Full FT | LoRA | Full FT | Full FT | LoRA |
| Additive | 2.81 | 7.40 | 3.50 | 13.33 | 15.57 | 6.12 | **2.81** | 5.83 | 7.01 | 10.02 | 38.90 | 21.22 |
| SVD | 2.80 | **7.14** | 3.48 | **10.41** | **13.74** | 5.89 | 2.82 | **5.76** | **6.80** | 7.03 | 40.47 | **20.23** |
| KNN | **2.79** | 7.33 | **3.45** | 12.80 | 14.30 | **5.54** | 2.84 | 5.83 | 6.83 | **7.71** | **38.77** | 20.82 |
| Similarity | 3.84 | 9.29 | 5.45 | 14.00 | 16.23 | 6.19 | 3.93 | 7.64 | 8.52 | 10.97 | 42.38 | 23.47 |

of-domain testing tasks. Details about the tasks included in the training, in-domain testing, and out-of-domain testing sets are discussed in Tables 7 and 8 in Appendix B.

To apply matrix completion for predicting forgetting, a few entries $z_{ij}$ should be known when a new fine-tuning task $T_i \in \mathcal{T}_{\text{test}}$ (row $i$) is introduced. We therefore assume access to the ground truth forgetting $z_{ij}$ of a tiny random set $\mathcal{S}$ ($|\mathcal{S}| = 30$) of upstream examples for $T_i \in \mathcal{T}_{\text{test}}$, noted as seed forgetting $z_i^{\mathcal{S}} = \{z_{ij}|x_j \in \mathcal{S}\}$. Obtaining seed forgetting typically takes only a few seconds by running inference with the model fine-tuned on $T_i$ over $\mathcal{S}$; we then predict forgetting of the rest $10k - 100k$ upstream examples. Figure 5 illustrates an example of the train-test partition, seed forgetting, and the forgetting to be predicted. We use Root Mean Squared Error (RMSE) over the $\mathcal{T}_{\text{test}}$ as the metrics of predicting example forgetting.

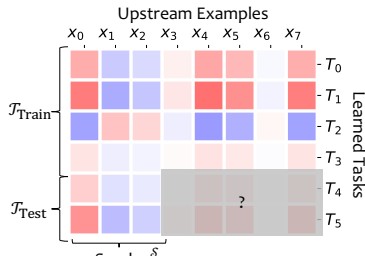

Figure 5: The training and testing setup of predicting example forgetting with association matrix completion.

**Matrix completion approaches**. We run matrix completion algorithms including additive linear, SVD, and k-nearest neighbors (KNN) models. The additive linear and the SVD models are introduced earlier in Sec. 3. Given the seed forgetting $z_i^{\mathcal{S}}$ of a task $T_i \in \mathcal{T}_{\text{test}}$, KNN finds tasks from $\mathcal{T}_{\text{train}}$ that have similar patterns of forgetting over the seed upstream examples $\mathcal{S}$. KNN computes an average of forgetting of top-k similar tasks from $\mathcal{T}_{\text{train}}$ weighted by their similarity as the prediction of forgetting caused by $T_i \in \mathcal{T}_{\text{test}}$ on the upstream examples $x_{1..M}$.

**Comparators of predicting forgetting.** We compare with a prior approach by Jin & Ren (2024) that leverages learned similarity between learned tasks and upstream examples by a trainable LM to predict forgetting. This prior work, however, focuses on predicting forgetting while fixing one single error in LM predictions; we extend the approach to predicting forgetting after fine-tuning models over a task (*i.e.,* a set of examples). The extended approach encodes upstream examples $x_j$ and the training examples of the learned task $x_i^{1..N_i} \in T_i$ with a trainable LM encoder $h(\cdot)$ to obtain their representations. The final prediction is made with a regression head over the inner products of two representations $\langle h(x_j), \frac{1}{N_i} \sum_{N_i} h(x_i) \rangle$.

We leave the implementation details of matrix completion approaches and the learned similarity approach in Appendix B.

### 4.2 MITIGATING FORGETTING WITH PREDICTED FORGETTING

**Leveraging predicted forgetting for mitigating forgetting**. We examine the practical utility of predicting forgetting as we sparsely replay upstream examples during forgetting following Jin & Ren (2024). Sparse replay of upstream examples is known as an effective and model-agnostic way to mitigate forgetting (de Masson D'Autume et al., 2019; Ibrahim et al., 2024). We replay one mini-batch of upstream examples every 32 training steps while fine-tuning on a new task. We perform targeted mitigation of forgetting by prioritizing examples that are predicted to suffer more from forgetting. This is achieved with weighted sampling of upstream examples $x_j$ proportional to $\exp(\hat{z}_{ij}/\tau)$, where $\hat{z}_{ij}$ are the predicted forgetting and $\tau$ is a temperature hyperparameter set as 0.1.

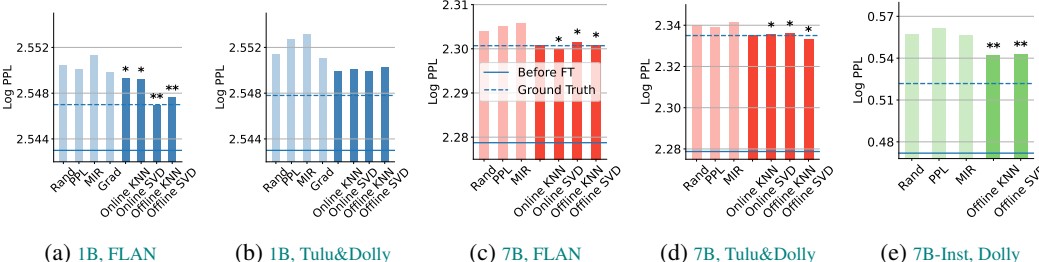

(a) 1B, FLAN  (b) 1B, Tulu&Dolly  (c) 7B, FLAN  (d) 7B, Tulu&Dolly  (e) 7B-Inst, Dolly

Figure 6: Log perplexity over upstream data as replay upstream examples selected by different approaches after fine-tuning OLMo or OLMo-Instruct models over in-domain or out-of-domain test tasks. The solid horizontal lines indicate the log perplexity before fine-tuning (*i.e.,* no forgetting). The dash lines show the log perplexity achieved by upweighting upstream examples according to their ground truth forgetting. * and ** indicate statistical significance of improvement ($p < 0.05$ or $p < 0.005$) compared to replaying random examples in paired $t$-tests over all fine-tuning tasks.

As we have discussed in Sec. 4.1, predicting forgetting with matrix completion requires seed forgetting $z^S$ to be evaluated. We consider an offline and an online variant of the approach. The *offline* variant performs a replay-free run of fine-tuning on the task $T_i$, after which the seed forgetting will be evaluated. We then perform another run of fine-tuning while replaying examples with the predicted forgetting. This creates computational overhead equivalent to one extra run of fine-tuning, but is still efficient when the training set of fine-tuning is considerably smaller than the upstream data. The *online* variant instead replays random examples for first 10% of fine-tuning steps, after which it evaluates seed forgetting and determine examples to be replayed in the rest of 90% steps. Compared to the offline variant, this mitigates the extra overhead of fine-tuning by trading off the prediction accuracy of forgetting.

**Baselines of mitigating forgetting.** We compare with diverse strategies of selecting upstream examples for sparse replay. We primarily examine whether weighted sampling with predicted forgetting statistically significantly improves over random sampling of upstream examples (Random). We also compare with Maximally Interfered Retrieval (MIR) (Aljundi et al., 2019a), a selection strategy sharing the similar notion of importance that forgotten examples should be selected for replay. The approach performs bi-level sampling by selecting the most forgotten examples from a small random subset of upstream data (set as $|\mathcal{S}| = 30$). In addition, we apply strategies that consider different definitions of upstream example importance. We examine an approach based on perplexity thresholds (PPL) (Marion et al., 2023), which samples upstream data of which the perplexity is around the median of the distribution. For OLMo-1B, we also sample replayed examples proportional to the gradient inner products (Grad-Prod) we evaluated in Sec. 3.3, in a similar vein to coreset selection approaches that utilize gradient information (Park et al., 2023; Xia et al., 2024). As a reference, we also experiment with upweighting upstream examples with ground truth forgetting $z_{ij}$, which, however, can face computational efficiency issues in practice.

**Metrics.** We measure log-perplexity increase over a held-out subset of 10,000 examples from the upstream data. This ensures none of the test examples are selected for replay by any of the example selection strategies.

### 4.3 RESULTS OF PREDICTING AND MITIGATING FORGETTING

**Results of predicting example forgetting**. Table 2 summarizes the error of predicting example forgetting over tasks from the in-domain and out-of-domain test splits. We see matrix completion approaches consistently outperform the learned similarity model in the prior work. Among the three matrix completion approaches, we notice that SVD models in general achieve the lowest prediction error. Besides, KNN in general outperforms additive linear and learned similarity models while being highly computationally efficient. Given this finding, we primarily leverage predictions given by the KNN and SVD models to reduce forgetting during fine-tuning.

**Mitigating forgetting with the predicted forgetting.** We leverage the online or offline predicted forgetting by the KNN and SVD models to reweight examples during replay following our procedure

in Sec. 4.2. Figure 6 summarizes log perplexity after fine-tuning over the held-out (never replayed) upstream data as we apply different upstream example selection approaches. We visualize the relative comparisons of these approaches to no forgetting of upstream examples (log perplexity before fine-tuning). We notice that example selection based on gradient inner products (Grad) or perplexity threshold (PPL), mainly applied for identifying important training data for a task in prior works, does not show improvement in mitigating forgetting compared to replaying random examples. This implies that the notions of example importance in these works are different from how easily the examples are forgotten. We also notice that MIR does not improve over random sampling in our setup, likely because of the small size of the retrieval candidates relative to the upstream examples. Upweighting examples with ground truth forgetting (GT) consistently reduces forgetting compared to random examples. By utilizing predicted forgetting by offline SVD and KNN we statistically significantly reduce forgetting compared to random examples in 4 out of 5 tested setups. The nuanced differences between SVD and KNN in forgetting mitigation align with their in-domain prediction accuracy of forgetting in Table 2, where SVD outperforms KNN by most on OLMo-7B models. Utilizing online-predicted forgetting also statistically significantly improves over replaying random examples in 3 of the setups. The gaps between online and offline variants are closer on 7B models than 1B models.

**Effects on downstream task performance measured with task-specific metrics.** We evaluate fine-tuned OLMo-7B or OLMo-7B-Instruct on unseen LLM leaderboard tasks and present the results in Appendix B. We notice the performance stays stable or improves on some downstream tasks while degrades on others, indicating forgetting. Although we observe slightly improved performance of Offline-KNN to random or no replay on most forgotten tasks (*e.g.*, Sciq on OLMo-7B and IFEval on OLMo-7B-Instruct), we do not see statistical significance. We leave effective algorithms to mitigate downstream task forgetting with predicted forgetting as future work.

**Computational Efficiency.** Table 3 summarizes the computation cost of the approaches as a function $FT(\cdot)$ of fine-tuning steps, a function $EV(\cdot)$ of upstream examples whose perplexity is evaluated, and the cost of matrix completion (MC) that is much smaller than LLM inference or training. We note the total number of upstream examples as $N$, the size of seed examples as $S$, and the number of fine-tuning steps as $Y$. As $S$ is much smaller than $M$, the majority of computational costs arise from fine-tuning $FT(Y)$ and $EV(N)$. Replaying with ground truth forgetting is the most costly, as it introduces an additional run of fine-tuning (after which forgetting will be evaluated) and infer-

Table 3: Computational cost of replay-based approaches as a summation of fine-tuning costs $FT(\cdot)$, inference costs over upstream examples $EV(\cdot)$, and matrix completion costs $MC$.

| Method | Cost |
|---|---|
| Random | $FT(Y)$ |
| Ground Truth | $2FT(Y) + EV(N)$ |
| Offline SVD, KNN | $2FT(Y) + EV(S) + MC$ |
| Online SVD, KNN | $FT(Y) + EV(S) + MC$ |
| MIR | $2FT(Y) + Y \cdot EV(S)$ |
| PPL,GradProd | $FT(Y)$ |

ence over potentially very large-scale upstream data. As offline prediction of forgetting only mitigates the need for inference, the approach saves computations when the cost of fine-tuning $FT(Y)$ is notably smaller than $EV(N)$, *i.e.,* over small fine-tuning datasets and massive upstream data. Online prediction of forgetting is always efficient, requiring only one run of fine-tuning and without the need to evaluate over massive upstream data.

# 5 RELATED WORKS

**Factors that affect forgetting**. In this paper, we primarily studied how the associations between learned and forgotten examples inform forgetting. Prior works have studied various factors that affect forgetting of the models, such as (1) type and size of the LM (Mehta et al., 2021; Scialom et al., 2022; Kalajdzievski, 2024; Mirzadeh et al., 2022) (2) trainable parts of the model (*e.g.*, LoRA, soft prompts, or full-model tuning) (Biderman et al., 2024a; Razdaibiedina et al., 2023) (3) hyperparameters such as learning rate (Ibrahim et al., 2024; Winata et al., 2023), dropout (Goodfellow et al., 2014), number of training steps (Biderman et al., 2024b; Kleiman et al., 2023) (4) optimizer (Lesort et al., 2023) and training algorithms (*e.g.*, various continual learning algorithms) (Shi et al., 2024; Wu et al., 2024), (5) the upstream examples or the knowledge themselves (Toneva et al., 2019; Zhang & Wu, 2024). Future works can study how various factors affect the associations between learned tasks and forgotten

examples. We consider empirical and theoretical study on the effect of task similarity on forgetting to be most relevant to ours. Ostapenko et al. (2022) empirically study relationships between task similarity and forgetting in foundation models over a sequence of newly learned tasks; our work instead focuses on forgetting of upstream data of LLMs. Theoretical study by Doan et al. (2020); Ding et al. (2024); Evron et al. (2022) dissects effects of the learned tasks on forgetting in linear models or around model initialization. We believe research on interpretations of forgetting (Tao et al., 2023; Zhao et al., 2023; Kotha et al., 2024) is complementary to ours and can potentially explain in the future why the associations in $Z$ are often simple, and in which circumstances the associations become more complicated.

**Data selection and data attribution.** Related to our work, data attribution studies faithful algorithms to find training examples that account for a prediction (Koh & Liang, 2017; Ilyas et al., 2022) from a pool of training examples. Park et al. (2023); Xia et al. (2024); Li et al. (2024); Liu et al. (2024) study the problem of selecting a subset of training data that maximizes performance on a given domain or task at a fixed budget for LLMs. Feldman & Zhang (2020); Tirumala et al. (2022); Biderman et al. (2024b); Swayamdipta et al. (2020) identify memorized, important, or forgetful training data. However, the notion of data importance in these works is different from how likely the upstream examples will be forgotten during fine-tuning. Furthermore, a systematical study on how such importance is dependent on newly learned tasks is still absent. Prior works represented by Aljundi et al. (2019a); Wang et al. (2024); Aljundi et al. (2019b) study selection strategies of examples for replay-based continual learning algorithms.

**Predicting model behaviors.** A number of works show LLMs can display a hybrid pattern of unpredictable to highly predictable behaviors (Ganguli et al., 2022; Wei et al., 2022). Ye et al. (2023); Xia et al. (2020); Schram et al. (2023) study prediction of task performance across datasets and training setups. We perform prediction at the example level which is more fine-grained and under-explored.

## 6 CONCLUSIONS

In this paper, we empirically analyzed the associations between learned and forgotten examples in LM fine-tuning. We showed the association displays a low rank pattern across different setups. We showed the example associations alone offer useful information to predict example forgetting when fine-tuning LMs on new tasks. We demonstrated the practical utility of our analysis by showing reduced forgetting as we reweight examples for replay with predicted forgetting. Future works can extend the study to a continual learning setup where new domains or tasks are sequentially learned while predicting forgetting of upstream examples online.

**Limitations.** In our research process, we tried to control confounding factors such as model size, learning rate, or the number of training steps. However, we did not systematically study how the associations between learned tasks and forgotten upstream examples depend on these factors. For example, although we presented results of 1B and 7B models from the same family (OLMo) and observe differences in their association statistics (such as $R^2$ scores in Sec. 3.1), it is unclear how the associations will appear in smaller or larger models. Besides, our empirical findings raise questions about interpretability of the associations, *i.e.,* why certain examples are more prone to forgetting while learning a new task, and whether there exist more human-interpretable patterns behind the associations. Finally, we limited our experiments to fine-tuning on a single task at a time. Predicting and mitigating forgetting in sequentially learned tasks is left as future work.

### REPRODUCIBILITY STATEMENT

All models and datasets used in our experiments are open-source with permissive licenses (see Appendix A for details). We will release code and statistics of forgetting collected in this study.

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

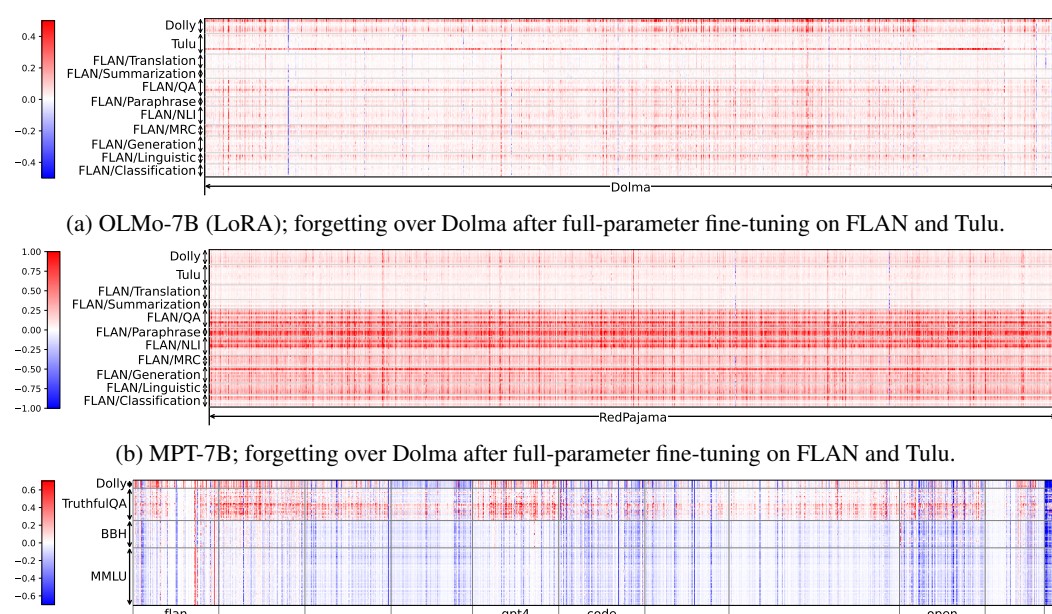

(a) OLMo-7B (LoRA); forgetting over Dolma after full-parameter fine-tuning on FLAN and Tulu.

(b) MPT-7B; forgetting over Dolma after full-parameter fine-tuning on FLAN and Tulu.

(c) OLMo-7B-Instruct (LoRA); forgetting over Tulu after full-parameter fine-tuning on unseen instruction-tuning tasks.

Figure 7: Additional visualized matrices of associations between learned tasks and forgotten examples. We plot forgetting (log-perplexity increase) that occurs on an upstream example (in $x$-axis) after learning a new task (in $y$-axis). Log-perplexity increase can be zero or negative, indicating no forgetting.

# A DATASET, MODEL, AND LM TRAINING DETAILS

**Models.** We use OLMo-7B[1] of the version pretrained on Dolma v1.6; and OLMo-7B-Instruct[2], which is tuned on Tulu v2 and other human feedback datasets.

**Learned new tasks and their categorization.** We summarize the list and the categorization of newly learned tasks in Tables 7 and 8 in our experiments. We also include the number of training examples and forgetting caused by each task averaged over all upstream examples.

**Training and evaluation details.** For full-parameter fine-tuning of OLMo-1B and 7B, we train the model for 1,000 steps with an effective batch size of 8 and a linearly decaying learning rate of $2e^{-6}$. For LoRA fine-tuning, we set the rank of adapters as 64 in all our experiments and use a rate of $10^{-4}$. We train the models for 625 steps with an effective batch size of 8. For OLMo-7B-Instruct and MMLU, BBH, TruthfulQA, considering the small size of the training sets, we train the models only for 37 steps with an effective batch size of 8. We use HuggingFace Transformers library for training and VLLM library for efficient inference. The statistics of forgetting are obtained in a single run.

**Dataset licenses.** MMLU and BBH are released under MIT license. Truthful QA, Dolma, Redpajama, OLMo models, and MPT models are released under Apache 2.0 license. Tulu V2 is released under ODC-By license. Dolly is released under CC BY-SA 3.0 license.

# B DETAILS OF FORGETTING PREDICTION AND REPLAY

**Data Splits for Predicting Example Forgetting.** We mark the tasks used as in-domain test splits for predicting example forgetting (Sec. 4) in Tables 7 and 8. The train-test split for the in-domain tasks is randomly generated.

---

[1]https://huggingface.co/allenai/OLMo-7B
[2]https://huggingface.co/allenai/OLMo-7B-Instruct

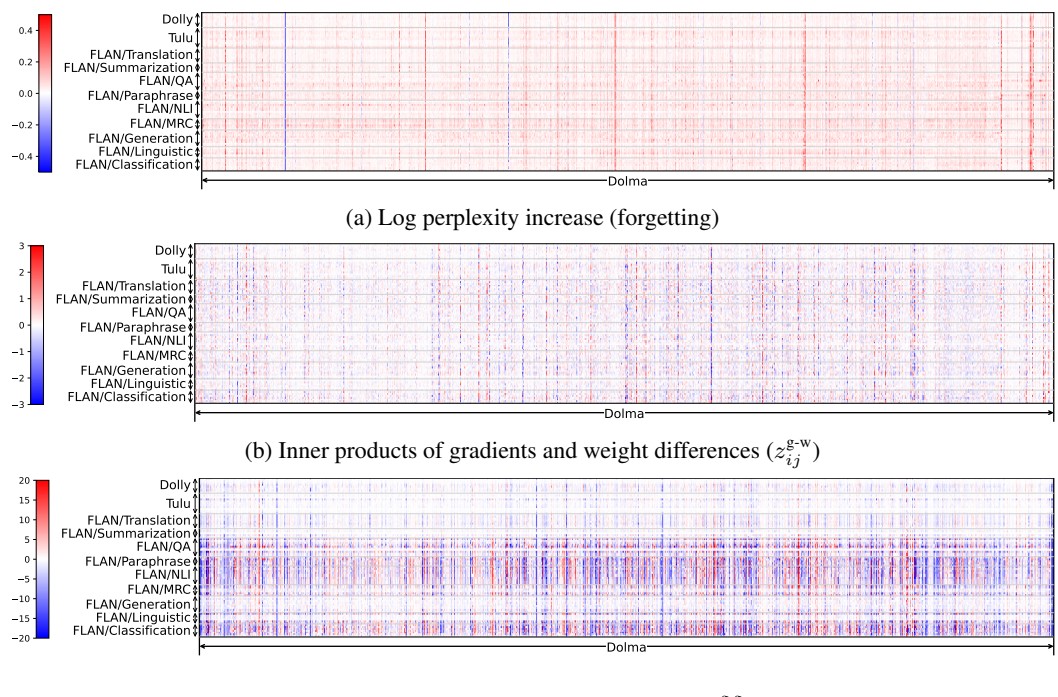

(a) Log perplexity increase (forgetting)

(b) Inner products of gradients and weight differences ($z_{ij}^{\text{g-w}}$)

(c) Negative inner products of gradients ($z_{ij}^{\text{g-g}}$)

Figure 8: A side-by-side comparison between the matrices of forgetting, inner products of gradients and weight differences ($z_{ij}^{\text{g-w}}$), and the negative inner products of gradients ($z_{ij}^{\text{g-g}}$) we examines in Sec. 3.3.

**Training and evaluation details.** We use Surprise Library 1.1.3[3] for additive linear, SVD, and KNN prediction models. For SVD, we set the dimension of the learnable features as 5. We train the regression models for 100 epochs over the association matrices. The other hyperparameters are left as default.

For in-domain test splits, we randomly sample 30 upstream examples and assume the ground truth forgetting is known for these examples. This is required for predicting forgetting on the rest of upstream examples by additive linear, SVD, and KNN methods. We repeat the experiment 10 times and report the mean and standard deviation in Table 2.

We used OLMo-1B models as the trainable example encoders in the implementation of the prediction method by Jin & Ren (2024) that relies on inner products of trained example representations. At inference, given an upstream example, we compute the averaged dot-product with all examples in the learned task. We note that at inference time the approach does not require ground truth forgetting of a small number of examples. For a fair comparison with other matrix completion methods, we replace the prediction of the approach with ground truth forgetting on these examples.

**Replaying upstream examples in fine-tuning.** We sparsely replay 1 mini-batch of 8 upstream examples every 32 steps of model update while fine-tuning on new tasks. Given predicted or ground truth forgetting $z_{i,1..J}$ on upstream examples $x_{1..J}$ when learning a new task $T_i$, we sample upstream examples to replay from a categorical distribution where $p(x_j) \propto \exp(z_{i,j}/\tau)$, where $\tau$ is a temperature hyperparameter set as 0.1. The hyperparameter $\tau$ is tuned on a single validation task by using ground truth forgetting $Z$.

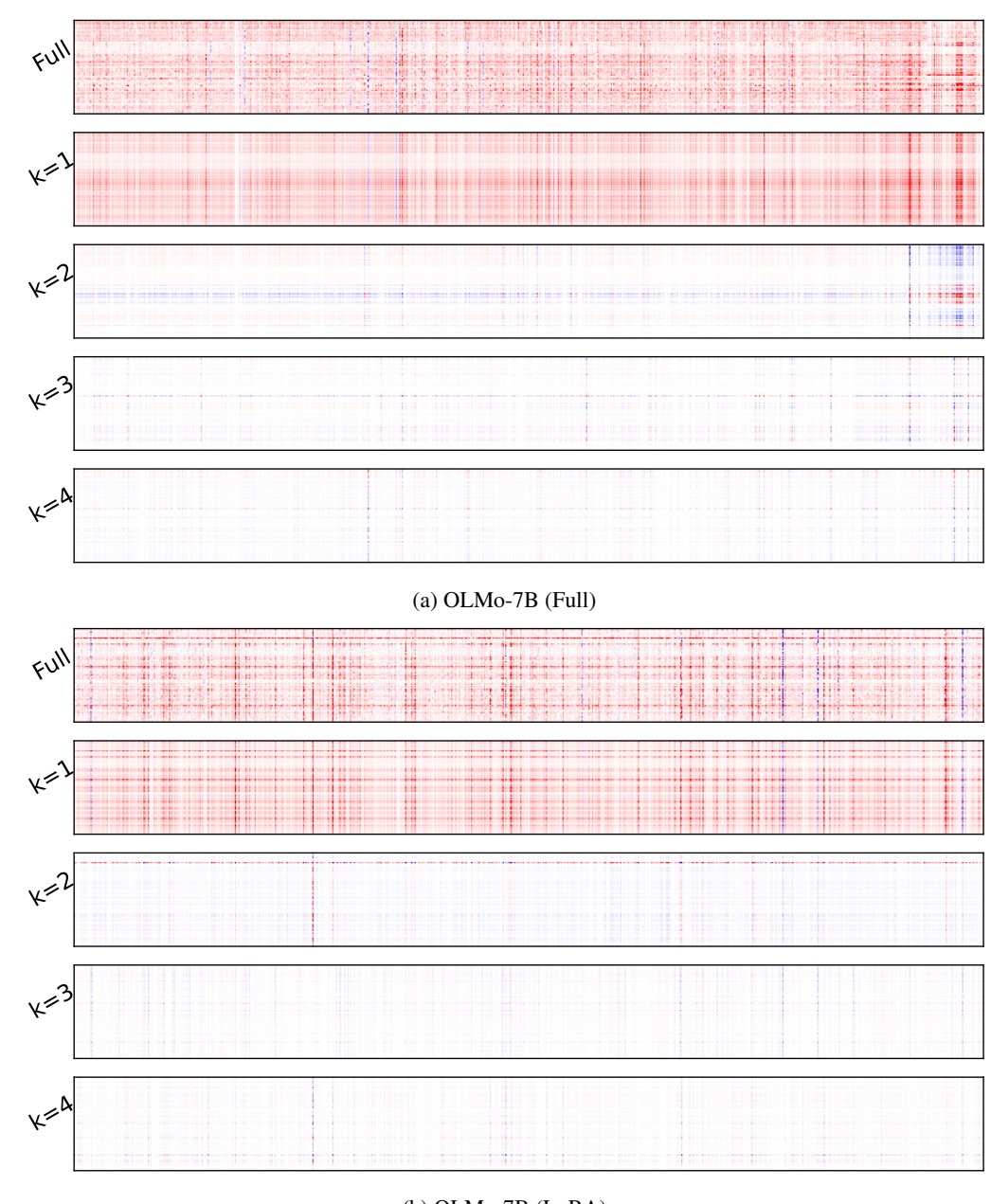

(a) OLMo-7B (Full)

(b) OLMo-7B (LoRA)

Figure 9: Reconstruction of $Z$ in OLMo-7B experiments with $k$-th singular value and vectors. Higher values of $k$ capture finer-grained details in $Z$.

Table 4: Downstream task performance of OLMo-7B models before and after fine-tuning on dolly tasks.

| | ARC-Easy | ARC-Challenge | Boolq | Hellaswag | Openbookqa | Piqa | Sciq | Winogrande |
|---|---|---|---|---|---|---|---|---|
| Metrics | Acc-norm | Acc-norm | Acc | Acc-norm | Acc-norm | Acc-norm | Acc-norm | Acc |
| Before FT | 68.77 | 40.36 | 72.41 | 75.65 | 42.20 | 79.54 | 88.60 | 66.29 |
| No Replay | 67.34 | 42.28 | 74.82 | 76.89 | 44.65 | 80.05 | 84.09 | 67.89 |
| Random | 67.48 | 42.43 | 74.33 | 77.26 | 44.88 | 79.97 | 84.77 | 67.33 |
| KNN-Offline | 67.49 | 42.24 | 74.33 | 77.07 | 44.30 | 80.09 | 84.91 | 67.54 |

Table 5: Downstream task performance of OLMo-7B-Instruct models before and after fine-tuning on dolly tasks.

| | MMLU-Pro | BBH | IF-Eval | MUSR | GPQA |
|---|---|---|---|---|---|
| Metrics | 5-shot Acc | 3-shot Acc-norm | 0-shot Inst | 0-shot Acc-norm | 0-shot Acc-norm |
| Before FT | 18.2 | 20.27 | 39.93 | 20.65 | 14.34 |
| No Replay | 17.51 | 20.26 | 17.72 | 22.01 | 14.37 |
| Random | 17.45 | 20.02 | 18.19 | 21.90 | 14.41 |
| KNN-Offline | 17.45 | 20.19 | 18.28 | 21.90 | 14.42 |

Table 6: Semantic meaning of $k$-th component in the SVD of the association matrix $Z$. We identify top relevant learned tasks and upstream example domains to $k$-th component in the SVD of the association matrix $Z$.

| | OLMo-1B | | OLMo-7B | |
|---|---|---|---|---|
| | Learned Tasks | More Forgotten Domain | Learned Tasks | More Forgotten Domain |
| $k = 1$ | flan/paws_wiki flan/glue_mrpc flan/story_cloze | None | flan/squad_v2 flan/fix_punct tulu/open_orca | None |
| $k = 2$ | flan/opinion_abstracts_idebate dolly/general_qa flan/story_cloze | StackOverflow | flan/mnli_matched flan/mnli_mismatched flan/snli | None |
| $k = 3$ | flan/story_cloze flan/fix_punct flan/true_case | None | flan/squad_v2 flan/quac flan/fix_punct | None |
| $k = 4$ | math_dataset dolly/general_qa flan/opinion_abstracts_idebate | None | flan/rte flan/opinion_abstracts_idebate flan/story_cloze | None |

# C DOWNSTREAM TASK EVALUATION WITH TASK-SPECIFIC METRICS

We evaluate downstream task performance of LMs before and after fine-tuning on 8 tasks from Dolly with LM-Evaluation-Harness framework (Gao et al., 2024). For OLMo-7B models, we evaluate on the same set of downstream tasks in OLMo technical report (Groeneveld et al., 2024). For OLMo-7B-Instruct models, we evaluate on Open LLM Leaderboard tasks[4]. For fine-tuned models, we compare no replay, replaying random examples, and replaying with forgetting predicted by offline KNN. Tables 4 and 5 summarize the results.

We notice that fine-tuning OLMo-7B on Dolly improves downstream performance on most of the downstream tasks. This aligns well with the purpose of fine-tuning a LM that is not instruction-tuned. Nevertheless, we notice performance degradation on two of the tasks, namely ARC-Easy and Sciq, which indicates forgetting. Although offline KNN achieves higher accuracy scores on these two tasks compared to no-replay (67.48 to 67.34, 84.91 to 84.09), we do not find the improvement statistically significant. For OLMo-7B-Instruct, fine-tuning on Dolly only improves performance on MUSR. The models clearly suffer from forgetting on the other tasks such as IFEval. Offline KNN achieves higher scores than random or no replay on IFEval (18.28 compared to 18.19 and 17.72), but we could not conclude about the significance of the improvement.

To summarize, we do not see clear performance improvement in downstream task performance (evaluated with task-specific metrics) by replaying random or chosen examples. We conjecture that replay-based approaches are not sufficient to mitigate forgetting on their own, and can be combined with other approaches such as careful learning rate scheduling or parameter regularization. We leave more effective algorithms to mitigate downstream task forgetting with predicted forgetting as future works.

## D  TOWARDS INTERPRETING FINE-GRAINED ASSOCIATIONS

We visualize progressive reconstruction with $k$-th singular value and singular vectors for OLMo experiments in Figure 9. The visualization exemplifies complicated associations that is not captured by the simple multiplicative model ($k = 1$). For example, on OLMo-7B (LoRA) and when $k = 2$, we see a single row and column with significantly larger forgetting than the others.

**Semantic meanings of $k$-th component in the SVD of the association matrix** $Z$. We perform further analysis into the patterns captured by the $k$-th singular value and singular vectors by identifying the most relevant learned tasks and upstream example domain to the component. For each $k$ and its corresponding component $\hat{Z}_k = s_k \boldsymbol{\alpha}_k \boldsymbol{\beta}_k^T$, we extract top 3 rows with the highest mean (*i.e.*, top 3 relevant learned tasks $T_i$). We also extract top 50 columns with highest mean (*i.e.* top 50 relevant upstream examples) and the domain where these upstream examples are drawn from. For OLMo models, the domains are one of C4, common-crawl, Gutenberg books, Reddit, Science, StackOverFlow, and Wikipedia. We compare the distribution of domains in the top 50 upstream examples, and perform a $z$-test to determine upstream example domain that is significantly more forgotten compared to a prior domain distribution of top 50 most forgotten upstream examples (colunms with highest mean in $Z$). The results are summarized in Table 6.

We highlight some notable patterns in Table 6. (1) Some component $Z_k$ highlights forgetting patterns of upstream examples from certain domains. On OLMo-1B, the second component ($k = 2$) highlights patterns where StackOverFlow examples are forgotten. (2) Some component $Z_k$ highlight forgetting when learning specific types of tasks. For example, the second component ($k = 2$) on OLMo-7B highlights forgetting patterns after learning NLI tasks (mnli_matched, mnli_mismatched, snli). This also exemplifies how learning similar tasks cause a similar set of upstream examples to be more forgotten. We believe a more comprehensive interpretation of the patterns of forgetting is an interesting and challenging future work.

---

[3] https://github.com/NicolasHug/Surprise/tree/v1.1.3
[4] https://huggingface.co/docs/leaderboards/open_llm_leaderboard/about

| Task Category | Task | Task Category | Task |
|---|---|---|---|
| FLAN/Classification | aeslc | FLAN/QA | arc_challenge* |
| | ag_news_subset | | arc_easy* |
| | imdb_reviews | | bool_q |
| | sentiment140 | | coqa* |
| | sst2 | | cosmos_qa |
| | trec* | | math_dataset* |
| | yelp_polarity_reviews* | | natural_questions* |
| FLAN/Linguistic | cola | | openbookqa* |
| | definite_pronoun_resolution* | | piqa |
| | fix_punct* | | trivia_qa* |
| | true_case | FLAN/Summarization | cnn_dailymail |
| | word_segment | | gigaword |
| | wsc* | | multi_news |
| FLAN/Generation | common_gen | | samsum |
| | copa | | wiki_lingua_english_en |
| | dart | FLAN/Translation | para_crawl_enes |
| | e2e_nlg* | | wmt14_enfr |
| | hellaswag | | wmt16_translate_csen |
| | opinion_abstracts_idebate* | | wmt16_translate_deen |
| | opinion_abstracts_rotten_tomatoes | | wmt16_translate_fien |
| | story_cloze | | wmt16_translate_roen |
| | web_nlg_en | | wmt16_translate_ruen* |
| FLAN/MRC | drop | | wmt16_translate_tren* |
| | multirc | Tulu | open_orca |
| | quac | | oasst1 |
| | record | | lima |
| | squad_v1 | | code_alpaca |
| | squad_v2 | | gpt4_alpaca |
| FLAN/NLI | anli_r1 | | cot |
| | anli_r2 | | science |
| | anli_r3 | | flan_v2 |
| | cb* | | sharegpt |
| | mnli_matched | | hard_coded |
| | mnli_mismatched | | wizardlm |
| | qnli* | Dolly | brainstorming |
| | rte | | closed_qa |
| | snli | | information_extraction |
| | wnli | | classification |
| FLAN/Paraphrase | glue_mrpc | | open_qa |
| | glue_qqp* | | general_qa |
| | paws_wiki | | creative_writing |
| | stsb | | summarization |
| | wic* | | |

Table 7: The list of learned tasks in our experiments on OLMo-1B, OLMo-7B and MPT-7B. * notes for tasks used as the in-domain test split in forgetting prediction experiments in Sec. 4.

| Task Category | Task | Task Category | Task |
|---|---|---|---|
| MMLU | abstract_algebra | BBH | boolean_expressions* |
| | anatomy | | causal_judgement |
| | astronomy | | date_understanding |
| | business_ethics | | disambiguation_qa |
| | clinical_knowledge | | dyck_languages* |
| | college_biology* | | formal_fallacies* |
| | college_chemistry | | geometric_shapes |
| | college_computer_science | | hyperbaton* |
| | college_mathematics | | logical_deduction_five_objects* |
| | college_medicine* | | logical_deduction_seven_objects |
| | college_physics | | logical_deduction_three_objects |
| | computer_security | | movie_recommendation* |
| | conceptual_physics* | | multistep_arithmetic_two |
| | econometrics | | navigate |
| | electrical_engineering | | object_counting* |
| | elementary_mathematics | | penguins_in_a_table |
| | formal_logic | | reasoning_about_colored_objects |
| | global_facts* | | ruin_names |
| | high_school_biology* | | salient_translation_error_detection |
| | high_school_chemistry | | snarks |
| | high_school_computer_science | | sports_understanding |
| | high_school_european_history* | | temporal_sequences |
| | high_school_geography | | tracking_shuffled_objects_five_objects |
| | high_school_government_and_politics | | tracking_shuffled_objects_seven_objects |
| | high_school_macroeconomics | | tracking_shuffled_objects_three_objects |
| | high_school_mathematics | | web_of_lies |
| | high_school_microeconomics | | word_sorting |
| | high_school_physics* | TruthfulQA | Nutrition |
| | high_school_psychology | | Stereotypes |
| | high_school_statistics | | Confusion |
| | high_school_us_history* | | Psychology |
| | high_school_world_history | | Language |
| | human_aging* | | Sociology |
| | human_sexuality* | | Finance |
| | international_law | | Indexical Error |
| | jurisprudence | | Science |
| | logical_fallacies* | | Misconceptions |
| | machine_learning | | Economics |
| | management* | | Education |
| | marketing* | | Proverbs |
| | medical_genetics | | Conspiracies |
| | miscellaneous | | Religion |
| | moral_disputes | | Statistics |
| | moral_scenarios* | | Misquotations |
| | nutrition | | Subjective |
| | philosophy* | | Law |
| | prehistory | | History |
| | professional_accounting | | Fiction |
| | professional_law | | Mandela Effect |
| | professional_medicine* | | Politics |
| | professional_psychology | | Misinformation |
| | public_relations* | | Logical Falsehood |
| | security_studies | | Distraction |
| | sociology* | | Weather |
| | us_foreign_policy* | | Myths and Fairytales |
| | virology | | Superstitions |
| | world_religions | | Advertising |
| | | | Paranormal |
| | | | Health |

Table 8: The list of learned tasks in our experiments on OLMo-7B-Instruct. * notes for tasks used as the in-domain test split in forgetting prediction experiments in Sec. 4.