# OpenReview forum: "Demystifying Language Model Forgetting with Low-Rank Example Associations"
_ICLR.cc/2025/Conference — Submitted to ICLR 2025_

### Official Review · Reviewer_d4Ds · 2024-10-31

**Soundness:** 2
**Presentation:** 4
**Contribution:** 2
**Rating:** 3
**Confidence:** 3

**Summary:**

This paper investigates forgetting in LLMs by building simple, low-rank models of example associations between upstream examples and downstream tasks. Its contributions are to
1. Demonstrate the simple low rank models can predict forgetting of upstream examples when models are trained on down stream tasks. It visualizes these associations and demonstrates goodness of fit. The authors also argue that these example association models are more predictive of forgetting than other baseline methods.
2. It predicts which examples will be forgotten when finetuned on new tasks and shows that this can be used to optimize the presentation of examples in a replay buffer to mitigate forgetting.

**Strengths:**

**Originality**: The paper takes an interesting and simple approach to understanding example forgetting and finds some intriguing results.
**Quality**: The paper follows a thread from understanding structure in these forgetting associations to developing a strategy to mitigate forgetting. It provides some baselines to argue for the effectiveness of all of its methods.
**Clarity**: The paper is very well written and clear. All the experiments, figures, methods, results are well explained and easy to follow.
**Significance**: The paper explores a relevant and important topic and presents an effective algorithm for mitigating forgetting.

**Weaknesses:**

1. While the fact that the forgetting association matrices are low rank, it is unclear what the takeaway from this is/how I should interpret or use this. Additionally I am not sure how to interpret any of the low rank structures. Also, the optimal predictor for forgetting with association matrix completion and what is used for forgetting mitigation is the kNN model which does not directly use this result?
2. I am not sure forgetting as measured by log perplexity is sufficiently convincing and would really like to see downstream accuracy on tasks. Is the forgetting that is observed actually hurting the capabilities of the models? Does it hurt the model's performance on benchmarks?

**Questions:**

1. What are the takeaways/what does it mean that forgetting association matrices are low rank? Should I expect them to be high rank or should that be the null hypothesis? For the multiplicative model, how should I interpret these R^2 values?
2. For the multiplicative model, do you have any insights on the $\alpha_i$ and $\beta_j$s? Which tasks cause more forgetting and why? Is it because their data is out of distribution? Amount of data? Which examples are more likely to be forgotten and why? Are they weird/noisy examples? Do they have higher loss compared to the mean across the dataset i.e. less well learned? Are they from a specific domain?
3. How do you interpret the fact that kNN does better than SVD at predicting forgetting?
4. Can you show that forgetting isn't just a perplexity phenomena and actually impacts performance on benchmarks? Can this benchmark performance degradation be predicted? The reason this is important is because it is possible to increase perplexity on certain examples on the training set, especially if they are noisy/rare examples without hurting language model capabilities and it would be useful to understand this.
5. It seems to me that predicting the association matrix is not really necessary for mitigating forgetting because the ground truth forgetting is easy to measure: you already have to fine-tune your model on the "test tasks" because you need to measure forgetting on the seed dataset. So you are just saving the inference cost on the rest of the upstream examples you are trying to not forget but just for the test tasks?

---

> ### Author Response · Authors · 2024-11-30
>
> We thank the reviewer for recognizing the originality and quality of our paper and the insightful questions. We address individual questions from the reviewer below.
>
> **Q1: For the multiplicative model, do you have any insights on $\alpha$ and $\beta$? Which tasks cause more forgetting and why?**
>
> We thank the reviewer for the insightful question. $\alpha$ indicates how much in general a learned task causes forgetting; $\beta$ indicates how much in general an upstream example suffers from forgetting. There exist prior works that focus on characterizing examples that are easily or never forgotten (high or low $\beta$) in classification tasks. For example, [1,2] suggests that rare (but helpful) or mislabeled examples are more likely forgotten, while typical / easy examples are hardly forgotten. However, as we focus on forgetting of instruction tuning and language modeling data, we find the forgotten examples difficult to characterize. We will include case studies in the final version.
>
> [1] Toneval et al. An empirical study of example forgetting during deep neural network learning, ICLR 2019
>
> [2] Maini et al. Characterizing datapoints via second-split forgetting, NeurIPS 2022
>
> **Q2: While the fact that the forgetting association matrices are low rank, it is unclear what the takeaway from this is/how I should interpret or use this.**
>
> We thank the reviewer for the insightful question. SVD approximates the forgetting association matrix with sum of rank-1 matrices. The good fit of rank-1 approximation implies that how easily an upstream example is forgotten is more of an intrinsic property of examples that is not relevant to the learned tasks. The second, third, and the rest of components in SVD indicate correlation between learned tasks and forgotten examples (that is less dominant). We updated Sec. 4.2 and Appendix D in the paper analyzing interpretable patterns in forgetting as indicated by each component in SVD.
>
> One takeaway of the low-rank approximation is that the patterns in forgetting are simple so that forgetting caused by unseen tasks can be well-predicted from statistics.
>
> **Q3. Should I expect forgetting associations to be high rank or should that be the null hypothesis? How should I interpret these R^2 values?**
>
> Theoretically grounded approximations of forgetting [1] and related works in linear models [2] suggest that forgetting depends on the similarity of two examples. It was however unclear how dominant the effect of example similarity is, and therefore the low rank approximation can be either good or bad. In this work, we demonstrate empirically that the association matrix $Z$ can be well-approximated with simple low rank models. For example, rank-3 approximation consistently yields R2 > 0.7. (70% of variance in forgetting can be explained by this low-rank approximation model).
>
> **Q4-1 : Can you show that forgetting isn't just a perplexity phenomena and actually impacts performance on benchmarks? Can this benchmark performance degradation be predicted?**
>
> We thank the reviewer for the suggestion.  We added additional experiment results evaluating fine-tuned models over LLM leaderboard tasks, also added to Sec. 4.3 and Appendix C. We see forgetting happens on downstream tasks as well.
>
> For OLMo-7B, we evaluated on the same set of downstream tasks in the OLMo technical report (ARC, Boolq, Hellaswag, Openbookqa, Piqa, Sciq, Winogrande) after fine-tuning on 8 dolly tasks. As expected, fine-tuning on Dolly improves accuracy on most downstream tasks, with exception on Sciq, where the accuracy decreased, indicating forgetting.
>
> Dataset | Before FT | No Replay | Random | KNN
> -- | -- | -- | -- | --
> Sciq | 88.60  | 84.09  | 84.77 | 84.91
>
> Similarly, for OLMo-7B-Instruct, we evaluate the model on MMLU-Pro, BBH, IFEval, MUSR, GPQA after fine-tuning on Dolly. We notice clear performance degradation on IFEval, where KNN reduces forgetting but not significantly.
>
> Dataset | Before FT | No Replay | Random | KNN
> -- | -- | -- | -- | --
> IFEval | 39.93 | 17.71 | 18.19 | 18.28
>
> We consider prediction of forgetting on benchmarks as an interesting future work that requires more time to experiment with.
>
> **Q4-2: …The reason this is important is because it is possible to increase perplexity on certain examples on the training set, especially if they are noisy/rare examples without hurting language model capabilities and it would be useful to understand this.**
>
> We totally agree with the reviewer that an additional process to filter out noisy, mistaken, or obsolete examples can be very beneficial and even necessary for continually updated LLM systems. We leave this research problem as a future work. In fact, we consider our analysis into forgetting may facilitate the research. We may, for example, prioritize upstream examples predicted to be forgotten when inspecting whether they are obsolete given the newly learned task.

---

> ### Author Response · Authors · 2024-11-30
>
> **Q5: How do you interpret the fact that kNN does better than SVD at predicting forgetting?**
>
> We appreciate the question from the reviewer. We performed an in-depth analysis into the performance issue of SVD. After extensive hyperparameter tuning, we notice that SVD could outperform KNN in predicting forgetting in a number of setups. We updated the results in Table 2, and also added results of replaying with SVD-predicted forgetting in Figure 6. We include the tuned results of SVD here.
>
> Method/Setup | OLMo-1B Full FT | OLMo-7B Full FT | MPT-7B | OLMO-7B-Inst Full FT
> -- | -- | -- | -- | --
> SVD | 2.80 | **7.14** | **10.41** | **13.74**
> KNN | **2.79** | 7.33 | 12.80 | 14.30
>
> We further notice that utilizing SVD-predicted forgetting slightly improves over KNN-predict forgetting in our example replay experiments (Figure 6) on OLMo-7B, where SVD clearly outperforms KNN in predicting forgetting.
>
>
> **Q6: It seems to me that predicting the association matrix is not really necessary for mitigating forgetting because the ground truth forgetting is easy to measure: you already have to fine-tune your model on the "test tasks" because you need to measure forgetting on the seed dataset.**
>
> We thank the reviewer for raising the issue. We added discussions about computation efficiency in Sec. 4.3 in the latest revision.
>
> To summarize,
>
> - Our previous approach enjoys most advantages when the upstream data is large-scale
> - In the latest version, we added a new variant of the approach that is always computationally more efficient than evaluating ground truth forgetting, that achieves statistically significant improvement in 3 out of 5 setups.

---

### Official Review · Reviewer_CsWA · 2024-10-31

**Soundness:** 1
**Presentation:** 1
**Contribution:** 2
**Rating:** 3
**Confidence:** 4

**Summary:**

This paper explores catastrophic forgetting of the pretraining or instruction tuning data after fine-tuning on downstream tasks. Specifically, the authors attempt to identify which samples (chunks of pretraining data) are forgotten when finetuning on various tasks, with a view to understanding the role of task similarity.

The authors analyse the amount of forgetting (measured by log perplexity increase) on each sample on a range of tasks. Analyzing the resulting data, the authors suggest that this data has a simple structure (though see weaknesses). This analysis relies upon a small selection of LLMs with open-source training data.

Building on this insight, the authors use matrix completion approaches predict which samples will be forgotten when training on a downstream task, though their approach requires access to forgetting data for some samples on the target task.

Finally, the authors use their prediction of which samples will be forgotten to upweight them during replay, and report improved forgetting mitigation compared to some other methods.

**Strengths:**

- The paper studies an important, timely problem, and connects previous research on task similarity to the contemporary LLM setting.
- Figure 1 is a helpful clarification of the authors' approach.
- The mitigation case study---using replay on their predicted forgotten samples---is a good idea to test the utility of their predictions.

**Weaknesses:**

- The results of this paper rest on the idea that that there is a simple structure that explains which samples will be forgotten when training on a given downstream task. However, I don't see clear evidence that this is the case. For example, referring to fig. 2, the authors state that they "see Z generally displays a neat and simple pattern." Such a pattern is not at all clear to me, and I would question the value of visualizing a large matrix in order to draw conclusions about the complexity of its structure.
- The authors rely on linear regression modelling to quantify how well forgetting is explained by (the indices of) each sample and task. The authors admission that "the R^2 scores are relatively lower [on some models]" makes me question the robustness of this approach. Why should the results vary so significantly between models of the same scale (e.g. OLMo-7B and MPT-7B) or even within the same family, where we can assume the pretraining data is the same (e.g. OLMo-7B and OLMo-1B)?
- What do the SVD components in section 3.2, fig. 4, tell us that we did not already know from visualizing the full matrix, e.g. in fig 2c? It is already evident from the full matrix that finetuning on TruthfulQA results in greater forgetting.
- The matrix completion approach requires access to the forgetting of some samples on the test tasks, significantly limiting its practical utility.
- The paper needs significant attention to improve its language and clarity. Numerous sentences don't make sense (e.g., "There has been a growing need for long-term usability of LLMs"), are ungrammatical (e.g., "We perform statistics of forgetting"), or are unclear (e.g., "As a main comparator"). I would also strongly encourage the authors to avoid usage of the vague term "associations" throughout the paper, where the authors tend to mean "task similarity" or "relationship between pretraining and fine-tuning task." For example, the "association matrix Z" might be better described as the "sample forgetting matrix Z".
- The paper lacks substantive discussion of its limitations, with the exception of lines 428-433 on the limitations of replay (rather than the limitations of their work more generally).
- **Minor**: It would be helpful to restructure the abstract to more clearly motivate the problem and what the authors propose. At present there excessive detail about constructing matrices, which makes it hard to understand that the paper's central concern is predicting catastrophic forgetting as a function of task similarity.

**Questions:**

1. Could the authors elaborate on why OLMo-7B and OLMo-7B-Instruct has "more complicated associations" than the similarly-sized MPT-7B (see weaknesses)? What is the intuition here?
2. The idea that the interaction between pretraining data and finetuning task determines forgetting would be supported by an additional baseline in the matrix completion experiments. Specifically, did the authors consider attempting matrix completion using just an average over finetuning tasks, rather than fitting to all tasks separately? Demonstrating that the average-over-tasks vector is insufficient would strengthen the results in table 2.
3. Could the authors provide more detail on how they determined their significance testing results in fig. 6? How did the authors correct for multiple comparisons?
4. For the replay mitigations in fig. 6, how was the kNN chosen rather than the additive linear or SVD methods (as mentioned on lines 347--353)? If these approaches were also tested, they should be reported, and need to be included in the multiple comparisons correction.
5. What are the fine-tuning tasks used for the mitigations experiments in section 4.2? This section is impossible to evaluate without this information.

---

> ### Author Response · Authors · 2024-11-30
>
> We thank the reviewer for recognizing the importance of the problem and the thoughtful comments. We address questions from the reviewer below.
>
> **Q1: The results of this paper rest on the idea that there is a simple structure that explains which samples will be forgotten when training on a given downstream task. However, I don't see clear evidence that this is the case.**
>
> We understand the confusion of the reviewer. To be precise, our paper tries to show that forgetting associations can be well-approximated with low-rank matrices. The majority of our results rely on quantitative measures such as R2 scores of approximations. We will revise imprecise statements like “Z generally displays a neat and simple pattern” in the final version of the paper.
>
> **Q2: The authors admit that "the R^2 scores are relatively lower [on some models]" makes me question the robustness of this approach. Why should the results vary so significantly between models of the same scale (e.g. OLMo-7B and MPT-7B) or even within the same family, where we can assume the pretraining data is the same (e.g. OLMo-7B and OLMo-1B)?**
>
> We thank the reviewer for this thoughtful question. Although R2 scores are lower on some models, the robustness of the predicting example forgetting and mitigating forgetting is validated across different experiment setups, including OLMo-7B, where the R2 scores are the lowest. For now, we could not provide a clear interpretation why the patterns of forgetting are different among models. Understanding the influence of these factors (model size, model family etc.) on patterns of forgetting may require significantly more experiments, which we leave as future work. We added discussions in the updated Limitations paragraph in Sec. 6.
>
> **Q3: What do the SVD components in section 3.2, fig. 4, tell us that we did not already know from visualizing the full matrix, e.g. in fig 2c?**
>
> We thank the reviewer for the thoughtful question. Though some patterns are obvious from the visualization, some more interesting patterns can be extracted by analyzing the decomposition with statistics or clustering. We demonstrated this for OLMo-1B and OLMo-7B models in Appendix C, and added pointers in Sec. 3.2.
>
> Some notable patterns include, for example,
> - On OLMo-1B, the second component (k=2) in SVD highlights forgetting where upstream examples from the StackOverFlow domains are disproportionately more forgotten, validated with a z-test. Top associated learned tasks are flan/opinion_abstracts_idebate, dolly/general_qa and flan/story_cloze.
> - On OLMo-7B, the second component in SVD highlights how NLI tasks (flan/mnli_mismatched, flan/mnli_matched, flan/snli) causes common upstream examples to be forgotten
> - Some components have much stronger correlation with TF-IDF textual similarity up to 0.33.

---

> ### Author Response · Authors · 2024-11-30
>
> **Q4: The paper lacks substantive discussion of its limitations**
>
> We thank the reviewer for the suggestion. We updated the limitations paragraph in Sec. 6, more specifically, there are questions yet to be answered:
>
> How does the complexity of associations depend on various factors such as model type or model size?
> What are more human interpretable patterns behind the forgetting?
> How can we extend the approach to a continual learning setup of multiple tasks?
>
> **Q5: For the replay mitigations in fig. 6, how was the kNN chosen rather than the additive linear or SVD methods? If these approaches were also tested, they should be reported.**
>
> We notice that, after extensive hyperparameter tuning, SVD could outperform KNN in predicting forgetting in many setups. We updated the results in Table 2, and also added results of replaying with SVD-predicted forgetting in Figure 6.
>
> We further notice that utilizing SVD-predicted forgetting slightly improves over KNN-predict forgetting in our example replay experiments (Figure 6) on OLMo-7B, where SVD outperforms KNN in predicting forgetting by a large margin.
>
> **Q6: Did the authors consider attempting matrix completion using just an average over finetuning tasks, rather than fitting to all tasks separately? Demonstrating that the average-over-tasks vector is insufficient would strengthen the results in table 2.**
>
> We consider this to resemble additive linear models without a learnable parameter for each fine-tuning task ($z_{ij}=b+\beta_j$, instead of $z_{ij}=b+\alpha_i + \beta_j$). We will add the results in the final version of the paper.
>
> **Q7: Could the authors provide more detail on how they determined their significance testing results in fig. 6?**
>
> We performed a paired samples t-test to determine whether replaying predicted examples outperforms random examples on 20 held-out tasks. The performance of replaying random or predicted examples on one single task forms a single “pair”.
>
> **Q8: What are the fine-tuning tasks used for the mitigation experiments in section 4.2?**
>
> We consider this to be a clarity issue, as we have marked the tasks used as training/testing sets in Tables 6 and 7 in Appendix and left pointers in the beginning of Sec. 4.1. We improved the clarity of the pointers in the revision.
>
> **Q9: Issues in language and writing**
>
> We sincerely appreciate the suggestions from the reviewer. We try to avoid too significant writing changes in this rebuttal revision for consistency. We will carefully revise the writing in the next version of the paper.

---

### Official Review · Reviewer_92Mo · 2024-11-01

**Soundness:** 4
**Presentation:** 3
**Contribution:** 2
**Rating:** 5
**Confidence:** 3

**Summary:**

The paper draws connections between upstream data that is forgotten during fine-tuning, and the new task.  It analyzes forgetting of past data after a model fine-tunes on new data using log perplexity, and shows how a matrix describing upstream loss given a new task can be described with low rank form. This matrix can then be used to predict forgetting of upstream tasks after fine-tuning. Finally, the paper describes a method to mitigate forgetting in fine-tuning by using weighted replay on upstream tasks that are more likely to be forgotten.

**Strengths:**

-I find it very interesting to describe forgetting using a low-rank representation. This shows that fine-tuning on tasks can be described using predictable forgetting patterns.

-It was very thought provoking to see how correlated the similarity of various tasks was to the likelihood of those tasks being forgotten.

-The paper provides a novel method to perform weighted replay for mitigating forgetting.

**Weaknesses:**

-Figure 2: it would be really helpful to add a description and possible analysis featuring what specific knowledge the examples cover, and thus the model forgets. Particularly if there is a correlation between forgetting and specific topics.

-The motivation for predicting forgetting rather than running inference could be more obvious: it is assumed that running inference on upstream examples is expensive, and thus matrix completion offers an alternative for predicting forgetting to mitigate it. However, given that there are enough resources available for a model to perform even costlier fine-tuning, it is not obvious why predicting forgetting rather than running inference is much more beneficial. Perhaps, a further explanation and a direct comparison of the approximate costs associated with performing matrix completion compared to running inference could be included.

-Figure 6: It would be beneficial to add another baseline which considers weighted sampling for replay based on the actual forgetting values for all data.

**Questions:**

-Figure 6: Random seems to outperform more complex baselines, can you explain why that might be?

---

> ### Author Response · Authors · 2024-11-30
>
> We thank the reviewer for finding the research interesting and the analysis thought-provoking. We address the questions from the reviewer below.
>
> **Q1: It would be really helpful to add a description and possible analysis featuring what specific knowledge the examples cover, and thus the model forgets. Particularly if there is a correlation between forgetting and specific topics.**
>
> We appreciate the suggestion from the reviewer. We updated Sec. 4.2 and Appendix D in the paper aiming at analyzing how each component in SVD corresponds to interpretable patterns in forgetting, i.e., how certain learned tasks cause specific types of upstream examples to be forgotten in OLMo-1B and OLMo-7B models.
>
> We summarize some interesting patterns, for example,
>
> - On OLMo-1B, the second component (k=2) in SVD highlights forgetting where upstream examples from the StackOverFlow domains are disproportionately more forgotten. Top associated learned tasks are flan/opinion_abstracts_idebate, dolly/general_qa and flan/story_cloze.
> - On OLMo-7B, the second component in SVD highlights how NLI tasks (flan/mnli_mismatched, flan/mnli_matched, flan/snli) causes common upstream examples to be forgotten
>
> At this stage, however, we consider leaving a more comprehensive semantic interpretation of patterns in forgetting as future work. This paper focuses on analyzing statistical properties of such patterns (e.g. complexity of associations by testing goodness-of-fit), and demonstrates how such statistical patterns enable better mitigation of forgetting. We added discussions about this future direction in the Limitations paragraph in Sec. 6.
>
> **Q2. It is not obvious why predicting forgetting rather than running inference is much more beneficial. Perhaps, a further explanation and a direct comparison of the approximate costs associated with performing matrix completion compared to running inference could be included.**
>
> We thank the reviewer for this question. We added discussion about computation efficiency in Sec. 4.3 in the latest revision.
>
> To summarize,
> - Our previous approach enjoys most advantages when the upstream data is large-scale
> - In the latest version, we added a new variant of the approach that is always computationally efficient while also performing competitively
>
> We include more details below.
>
> We let $FT(n)$ be the cost of fine-tuning on n examples, and $EV(m)$ be the cost of evaluating on m upstream examples, and MC be the cost of matrix completion algorithms. We let X and Y be the number of upstream examples and fine-tuning steps, and S be “seed” upstream examples for matrix completion where S << X (Sec. 4.1). The computational costs can be summarized as follows.
>
> Method | Computational Cost
> -- | --
> Random replay | FT(Y)
> Replay with ground truth forgetting | 2FT(Y) + EV(X)
> Replay with offline predicted forgetting | 2FT(Y) + (EV(S) + MC) (small)
> Replay with online predicted forgetting | FT(Y) + (EV(S) +  MC) (small)
> Maximally Interfered Retrieval (MIR) baseline | 2FT(Y) + Y EV(S)
>
> \* We added the “online” variant of forgetting prediction in our latest revision, which we detailed in Sec. 4.2.
>
> We understand the question from the reviewer about how the cost of fine-tuning (FT(Y)) compares to inference on upstream data (EV(X)). This determines how much computation can be saved via (offline) prediction of forgetting, compared to directly evaluating ground truth. In fact, we discussed the limitation (previously in the end of Sec. 4.3) and showed that the computation can be saved when the upstream data is large. This was the case in our OLMo-1B and OLMo-7B experiments, where we fine-tuned the model for 1,000 steps (with batch size 8), while there were 140,186 upstream examples.
>
> Nevertheless, the online variant of our approach added in the latest revision is much more efficient than replaying with ground truth forgetting in any scenarios. The variants trades off accuracy in predicting forgetting for computational efficiency, and also achieves significant improvement compared to random replay in 3 of the 5 setups, as we discussed in the updated Sec. 4.3.

---

> > ### Author Response · Authors · 2024-11-30
> >
> > **Q3: It would be beneficial to add another baseline which considers weighted sampling for replay based on the actual forgetting values for all data.**
> >
> > We consider this to be a clarity issue, as we have already included the results of replaying with ground truth forgetting value in Figure 6 (in the dashed line, as a potential upper bound of replaying with predicted forgetting) and discussed the results in Sec. 4.3. In the latest revision, we added more pointers at the end of the paragraph “Baselines of mitigating forgetting” in Sec. 4.2.
> >
> > **Q4:  Random seems to outperform more complex baselines, can you explain why that might be?**
> >
> > We thank the reviewer for being careful. We discussed possible reasons in Sec. 4.3. Most of these example selection strategies are proposed to identify “important” upstream examples to certain criteria. However, our results show that such “importance” in prior works do not reflect how such upstream examples are useful for mitigating forgetting.

---

### Official Review · Reviewer_YyNk · 2024-11-03

**Soundness:** 2
**Presentation:** 3
**Contribution:** 2
**Rating:** 6
**Confidence:** 3

**Summary:**

The paper investigates the effect of instruction-tuning on language model forgetting and the strategies to mitigate them. In particular, the paper discovers that the associations between learned tasks and forgotten examples can be reasonably described by simple linear/multiplicative models, exhibiting low-rank patterns. This association can be used to select replay upstream samples when training the model on downstream tasks to mitigate the forgetting of prior knowledge.

**Strengths:**

1. The paper is well written with notable clarity.
2. The paper provides extensive and well-designed quantitative and qualitative studies to understand the correlation between forgetting the upstream knowledge and tuning on the downstream tasks.
3. The paper highlights that the importance/similarity of the examples to the target downstream tasks is a different notion compared to their sensitivity to be forgotten.
4. The proposed predict-then-upweight replay strategy effectively mitigates the sample forgetting for upstream pretraining/instruction-tuning data in terms of perplexity.

**Weaknesses:**

1. Reducing LLM forgetting of the upstream knowledge during instruction-tuning and alignment stages is an important aspect, but typical instruction-tuning and alignment datasets have other metrics for evaluation. It is unclear whether reducing forgetting has practical benefits or trade-offs to these downstream tasks.
2. There are extensive discussions about using SVD to approximate the matrix, which certainly proves the hypothesis that leveraging a simple forgetting predictor can be more helpful than task-related features. However, the final forgetting predictor using KNN over SVD reduces the importance of those discussions.

**Questions:**

1. Could you provide more justifications on why SVD reveals fine-grained associations? Since increasing the rank $r$ of SVD provides a better fitting of the forgetting (perplexity) matrix, different SVD components ($r$-th component) may have opposite patterns since only the summation of all components matters.
2. Are there particular reasons that the Spearman correlation is chosen over the Pearson correlation in Section 3.3? In addition, have you experimented with the correlation between forgetting and the cosine similarity of the textual representations (e.g., last token representation), which is an alternative to TF-IDF for textual similarity?
3. In Table 2, it seems that forgetting is more difficult to predict for instruction-tuned OLMo-7B-Instruct. Why is that so? Is it correlated with the absolute perplexity of the upstream data?
4. As also raised in weakness 1, how does the proposed forgetting-mitigation approach affect the performance on the downstream tasks (or even the upstream tasks) that are typically not evaluated by perplexity?

---

> ### Author Response · Authors · 2024-11-30
>
> We thank the reviewer for recognizing the strengths of our paper that the topic is interesting and well-motivated. We address the questions from the reviewer below.
>
> **Q1. The final forgetting predictor using KNN over SVD reduces the importance of those discussions.**
>
> We appreciate the question from the reviewer. Following the question, we performed an in-depth analysis into the performance issue of SVD. After extensive hyperparameter tuning, we notice that SVD could outperform KNN in predicting forgetting in a number of setups. We updated the results in Table 2, and also added results of replaying with SVD-predicted forgetting in Figure 6. We include the tuned results of SVD here.
>
> Method/Setup | OLMo-1B Full FT | OLMo-7B Full FT | MPT-7B | OLMO-7B-Inst Full FT
> -- | -- | -- | -- | --
> SVD | 2.80 | **7.14** | **10.41** | **13.74**
> KNN | **2.79** | 7.33 | 12.80 | 14.30
>
> Our analysis in Sec. 3.1 and Sec. 3.2 with SVD try to analyze how complicated are the associations between learned tasks and forgotten examples. Like many other application scenarios of matrix completion, we believe simple or tractable patterns in Z is a necessary condition for these algorithms to succeed, which is not restricted to SVD. Following this, we believe there exist untested matrix completion algorithms that perform better. We will include more matrix completion algorithms in the final version as possible.
>
> **Q2. It is unclear whether reducing forgetting has practical benefits or trade-offs to these downstream tasks (evaluated with task-specific metrics) / How does the proposed forgetting-mitigation approach affect the performance on the downstream tasks (or even the upstream tasks) that are typically not evaluated by perplexity?**
>
> We thank the reviewer for this thoughtful question. We added additional experiment results evaluating fine-tuned models over LLM leaderboard tasks. These results are discussed in updated Sec. 4.3, and Appendix C.
>
> For OLMo-7B, we evaluated on the same set of downstream tasks in the OLMo technical report (ARC, Boolq, Hellaswag, Openbookqa, Piqa, Sciq, Winogrande) after fine-tuning on 8 dolly tasks. As expected, fine-tuning on Dolly improves accuracy on most downstream tasks, with exception on Sciq, where the accuracy decreased (indicating forgetting).
>
> Dataset | Before FT | No Replay | Random | KNN
> -- | -- | -- | -- | --
> Sciq | 88.60  | 84.09  | 84.77 | 84.91
>
> We notice replaying examples predicted by KNN reduces forgetting, but we could not make claims about statistical significance.
>
> Similarly, for OLMo-7B-Instruct, we evaluate the model on MMLU-Pro, BBH, IFEval, MUSR, GPQA after fine-tuning on Dolly. We notice clear performance degradation on IFEval, where KNN reduces forgetting but not significantly.
>
> Dataset | Before FT | No Replay | Random | KNN
> -- | -- | -- | -- | --
> IFEval | 39.93 | 17.71 | 18.19 | 18.28
>
> We include full results on other downstream tasks in Tables 3 and 4 in Appendix C.
>
> To summarize, although we observe statistically significantly reduced forgetting on upstream instruction tuning or language modeling data evaluated with perplexity, we do not observe significant improvement in downstream tasks evaluated with task-specific metrics. We consider improving downstream task performance with predicted forgetting is a valuable future work.

---

> ### Author Response · Authors · 2024-11-30
>
> **Q3: Are there particular reasons that the Spearman correlation is chosen over the Pearson correlation in Section 3.3? In addition, have you experimented with the correlation between forgetting and the cosine similarity of the textual representations (e.g., last token representation), which is an alternative to TF-IDF for textual similarity?**
>
> We thank the reviewer for the suggestion. Our earlier consideration was that Pearson correlation implicitly assumes linear relationship, which might not be the case. Nevertheless, in the new version, we reported both Pearson and Spearman’s correlation in Table 1 for completeness of the results. We see the trends are consistent between two types of correlations.
>
> Additionally, we evaluated correlations between forgetting and cosine similarity of last layer token representation of OLMo-1B models.
>
>  Pearson $\rho$ | Spearman $\rho$
> -- | --
>  0.021  | 0.017
>
> The results are added to Table 1. Just as other similarity measures, we see a low correlation.
>
> **Q4: Since increasing the rank of SVD provides a better fitting of the forgetting (perplexity) matrix, different SVD components (k-th component) may have opposite patterns since only the summation of all components matters?**
>
> We understand the concern from the reviewer. However, one property of SVD (as in L239) is that rank-r reconstruction is the optimal for minimizing the reconstruction error $|| Z - Z_r ||_F$. Therefore, although there could be conflicting signs in different components, there will not be cases where two components completely cancel out. Nevertheless, we consider non-negative matrix factorization may be of interest to the reviewer and future readers, where all components are positive and thus more interpretable. We will add the results in the final version of the paper.
>
> We also added more analysis into patterns captured by SVD in Appendix D.
>
> **Q5: Table 2, it seems that forgetting is more difficult to predict for instruction-tuned OLMo-7B-Instruct. Why is that so?**
>
> We thank the reviewer for the thoughtful question. At this stage, we plan to leave how various factors influence the complexity of the associations as a future work. We included the discussion in the updated Limitations paragraph in Sec. 6.

---

### Official Review · Reviewer_YpEh · 2024-11-04

**Soundness:** 2
**Presentation:** 3
**Contribution:** 1
**Rating:** 3
**Confidence:** 3

**Summary:**

The paper investigates fine-grained associations between new tasks learned by LLMs and catastrophic forgetting of upstream data previously used for pretraining, under the lens of matrix completion. Modeling LLM forgetting as an $M\times N$ matrix with $M$ new tasks and $N$ upstream datapoints, the authors discover that the associations exhibit a low-rank pattern as a simple multiplicative regression model such as low-rank SVD provide a fairly accurate estimation of the matrix. Inspired by this finding, the paper proposes a novel example replay framework that replays upstream examples predicted to be forgotten under matrix completion during finetuning, thereby mitigating catastrophic forgetting.

**Strengths:**

- [S1] **Interesting problem and motivation.** Considering wide use of LLM adaptation, understanding fine-grained associations between downstream tasks vs. catastrophic forgetting has great potential as claimed by the authors, as it can inspire techniques for better knowledge retention.
- [S2] **Great presentation.** The paper presents interesting analyses and comparisons with previous similarity-based measures, making it easy to understand comprehensively understand the motivation and inspirations of this work.

**Weaknesses:**

Despite its strengths, the paper is weak with respect to practicality and experimental results, details of which are as follows.

- [W1] **Weak practicality under large number of downstream tasks.** The proposed framework requires finetuning the base LLM on individual tasks to gather train data for matrix completion, which incurs a high computational cost.  Considering unprecedented model sizes as well as range of downstream tasks for LLMs, we can expect the cost to quickly become intractable in many scenarios. This could be alleviated if we can instead sample a small number of finetuning tasks, and show that the proposed method is robust under small choices of such tasks, but this is somewhat unclear in the current draft.

- [W2] **Weak practicality due to requiring full access to upstream data.** The proposed example replay framework requires full access to upstream data during finetuning, yet we may no longer have access to exact upstream data points due to updates on the data or data sharing policies [A]. Ideally, it would be best to mitigate forgetting on upstream examples without explicitly using them during fine-tuning, yet the matrix completion formulation used in this work seems unsuitable for that direction.

- [W3] **Unconvincing experimental setup.** The proposed framework makes two runs of finetuning, one without replays to gather forgetting information $z_{ij}$ on the train set and perform matrix completion to pick upstream examples to replay, then a second finetuning of the model with intermediate replays on examples chosen. Given this process, it is unclear whether the statistics gathered *post-finetuning without any replay* will hold at every step of *finetuning with replay*, and additional analyses on the predicted forgetting without replay vs. with replay or a similar experiment seems necessary.

- [W4] **Insufficient performance gains.** While the main results in Figure 6 show that the proposed KNN-based approach is statistically better than the random baseline, all methods are significantly far from pre-finetuning performances, leading to concerns on the effectiveness of replay-based methods in mitigating LLM forgetting overall, which weakens the proposed method's practical utility claimed by the authors.

[A] Gao et al., The Pile: An 800GB Dataset of Diverse Text for Language Modeling

**Questions:**

Please refer to concerns in the weaknesses block above.

---

> ### Author Response · Authors · 2024-11-30
>
> We thank the reviewer for recognizing the strengths of our paper that the topic is interesting and well-motivated. We address the questions from the reviewer below.
>
> **Q1.  Weak practicality under a large number of downstream tasks.**
>
> We agree with the reviewer that obtaining ground truth forgetting of upstream examples is computationally expensive. However, collecting training data is a one-time effort that allows forgetting of new tasks (from the test set) to be predicted efficiently. As we perform more runs of fine-tuning over new tasks or incrementally collected datasets, we may eventually reach a point where the overhead of collecting training data becomes worthy. In fact, our research into predicting example forgetting is motivated by reducing the computational cost of repetitively evaluating forgetting in large-scale scenarios, when LMs are fine-tuned for a diverse set of tasks.
>
> We added discussions about the computational efficiency after Sec. 4.3 in the revision.
>
> **Q2.  Weak practicality due to requiring full access to upstream data.**
>
> We understand the concern from the reviewer. It seems the limitation roots in replay-based algorithms to mitigate forgetting. Still, we focused on replay-based algorithms, because of advantages such as being scalable and easy to implement when data availability is less of an issue.
>
> **Q3.  It is unclear whether the statistics gathered post-fine tuning without any replay will hold at every step of fine tuning with replay**
>
> We thank the reviewer for the insightful question. We measure log-perplexity of replayed and never-replayed upstream examples separately after fine-tuning the LM while replaying random examples.
>
> For those very few replayed upstream examples, we notice the perplexity drops significantly as expected; for the majority of upstream examples that are never replayed, the perplexity decreases on average (which indicates reduced forgetting. Nevertheless, we find the perplexity of individual upstream examples with replay strongly correlates with their perplexity without any replay (0.98 Spearman correlation on OLMo-7B). Therefore, we consider the statistics gathered is still very informative regardless whether or not replay is performed. We leave more in-depth analysis as future work.
>
> **Q4. Insufficient performance gains of replay-based approaches**
>
> We thank the reviewer for pointing the issue out. Our main goal is to demonstrate the benefits of predicting example forgetting, as exemplified in replay-based methods. We are aware of alternative approaches to mitigate forgetting with a subset of training examples [1,2], for example, regularizing parameter updates or applying knowledge distillation, where predicting forgetting can play a role. We leave integration of predicted forgetting into these approaches as future work.
>
> [1] Nguyen et al. Variational Continual Learning, ICLR 2018
>
> [2] Li et al. Learning without Forgetting, ECCV 2016

---

### Author Response · Authors · 2024-11-30
**General response to reviewers**

We thank all the reviewers for their thoughtful and constructive comments. We are grateful for reviewers recognizing the strengths of our paper that analyzing and predicting forgetting is an interesting and well-motivated topic. We made updates to the paper to address common questions from the reviewers, highlighted with colored text.

1. **Performance issues of SVD [YyNk, CsWA, d4Ds].** We revised our results of SVD and noticed that SVD could outperform KNN after extensive hyperparameter tuning and longer training runs. We updated Table 2 of predicting example forgetting accordingly. We also added SVD in our example replay experiments and presented them in Figure 6.

2. **Computational efficiency [92Mo, d4Ds].** We added a new paragraph in Sec. 4.3 to discuss computational efficiency. We also added a new efficient “online” variant of forgetting prediction in example replay experiments that performs competitively, without extra fine-tuning overhead. The variant is introduced in Sec. 4.2, and we presented results in Figure 6.

3. **Semantic interpretation of associations between learned tasks & forgotten examples [92Mo, CsWA, dsDs].** We present how SVD reveals fine-grained interpretable association in OLMo-1B and OLMo-7B in Appendix D, complementary to our previous analysis about OLMo-7B-Instruct in Sec. 3.2. We leave more comprehensive interpretations as future works in the updated Limitations paragraph in Sec. 6.
4. **Effect on downstream task performance evaluated with task specific metrics [YyNK, d4Ds].** We added evaluation on LLM benchmark datasets of fine-tuned models with different replay strategies. We notice although replaying with predicted forgetting reduces forgetting on most forgotten tasks in numbers, the improvements were not statistically significant enough.
5. We fixed errors in some numbers in Table 1 “Correlations between various measures of similarity and the actual forgetting”

---

### Meta-Review · Area_Chair_aFth · 2024-12-19

**Metareview:**

This work investigates the fine-grained associations between new tasks learned by LLMs and the phenomenon of catastrophic forgetting of upstream pretraining data. Through the perspective of matrix completion, the study models LLM forgetting as an $M \times N$ matrix, where M represents new tasks and N represents upstream data points. The authors find that these associations exhibit a low-rank structure, as simple multiplicative regression models, such as low-rank SVD, provide a reasonably accurate estimation of the matrix. Building on this observation, the paper introduces a novel example replay framework that mitigates catastrophic forgetting by replaying upstream examples predicted to be forgotten under the matrix completion framework during fine-tuning.

Following discussions, the reviewers reached a consensus to reject this work. The primary concerns raised include the limited practicality of the approach due to the reliance on obtaining ground truth forgetting data, insufficient performance gains, a lack of in-depth analysis, insufficient experimental validation of the main claims, and various other experimental shortcomings. As a result, the current version does not appear ready for publication. We recommend that the authors address these concerns thoroughly in line with the reviewers' feedback to strengthen the work.

**Additional Comments On Reviewer Discussion:**

I mainly list the key concerns since different reviewers have different concerns.

1)	limited practicality under a large number of downstream tasks (reviwer YpEh)
The authors agree that obtaining ground truth forgetting of upstream examples is computationally expensive, but explain  collecting training data is a one-time effort that allows forgetting of new tasks (from the test set) to be predicted efficiently.

2)	limited practicality due to the need for obtaining ground truth forgetting data (Reviewers YpEh).
It seems the limitation roots in replay-based algorithms to mitigate forgetting. Still, we focused on replay-based algorithms, because of advantages such as being scalable and easy to implement when data availability is less of an issue.

3)	insufficient performance gains (reviwer d4Ds)
The authors do not provide extra experimental results, and only claim their target and contribution.

4)	lacking depth analysis (reviwer YpEh, 92Mo)
e.g. 1) While the fact that the forgetting association matrices are low rank, it is unclear what the takeaway from this is/how I should interpret or use this. 2) For the multiplicative model, do you have any insights on α and β? Which tasks cause more forgetting and why?
The authors provide some explanations that but are not so convincing.

5)	Unclear experimental setup and explain (reviewer CsWA,  YpEh, 92Mo)
The authors provide some explanations that but are not so convincing.

---

### Decision · Program_Chairs · 2025-01-22

Reject